# Enhanced and Interpretable Prediction of Multiple Cancer Types Using a Stacking Ensemble Approach with SHAP Analysis

**DOI:** 10.3390/bioengineering12050472

**Published:** 2025-04-29

**Authors:** Shahid Mohammad Ganie, Pijush Kanti Dutta Pramanik, Zhongming Zhao

**Affiliations:** 1AI Research Centre, Department of Analytics, Woxsen University, Hyderabad 502345, India; shahidmohd728@gmail.com; 2School of Computer Science and Engineering, Galgotias University, Greater Noida 203201, India; 3Center for Precision Health, McWilliams School of Biomedical Informatics, The University of Texas Health Science Center at Houston, Houston, TX 77030, USA

**Keywords:** cancer prediction, ensemble learning, stacking, lung cancer, breast cancer, cervical cancer, XAI, SHAP

## Abstract

**Background**: Cancer is a leading cause of death worldwide, and its early detection is crucial for improving patient outcomes. This study aimed to develop and evaluate ensemble learning models, specifically stacking, for the accurate prediction of lung, breast, and cervical cancers using lifestyle and clinical data. **Methods**: 12 base learners were trained on datasets for lung, breast, and cervical cancer. Stacking ensemble models were then developed using these base learners. The models were evaluated for accuracy, precision, recall, F1-score, AUC-ROC, MCC, and kappa. An explainable AI technique, SHAP, was used to interpret model predictions. **Results**: The stacking ensemble model outperformed individual base learners across all three cancer types. On average, for three cancer datasets, it achieved 99.28% accuracy, 99.55% precision, 97.56% recall, and 98.49% F1-score. A similar high performance was observed in terms of AUC, Kappa, and MCC. The SHAP analysis revealed the most influential features for each cancer type, e.g., fatigue and alcohol consumption for lung cancer, worst concave points, mean concave points, and worst perimeter for breast cancer and Schiller test for cervical cancer. **Conclusions**: The stacking-based multi-cancer prediction model demonstrated superior accuracy and interpretability compared with traditional models. Combining diverse base learners with explainable AI offers predictive power and transparency in clinical applications. Key demographic and clinical features driving cancer risk were also identified. Further research should validate the model on more diverse populations and cancer types.

## 1. Introduction

Modern lifestyle has contributed to the emergence of numerous critical and life-threatening diseases. Among these, cancer stands out as the leading cause of death, responsible for nearly 10 million fatalities worldwide in 2020, equating to one in every six deaths [1]. Cancer presents significant challenges in diagnosis, treatment, and prognosis. Accurate and early cancer diagnosis improves patient outcomes and reduces healthcare costs [2]. Although advancements in medical imaging, genomics, and biomarker discovery have led to improved cancer detection, there is still a need for more accurate and reliable diagnostic tools, especially for complex cancers with heterogeneous presentations. In this context, the development of machine learning models for predicting different types of cancer has gained significant attention [3,4].

Traditional machine-learning models face several limitations in cancer prediction [5]. One of the key challenges is limited feature selection, as cancer is a complex disease influenced by a multitude of factors, including age, gender, lifestyle, genetics, and environmental factors. Identifying the most relevant features that can effectively predict the occurrence of cancer poses a significant challenge for conventional machine learning algorithms. Another limitation is the issue of imbalanced data, in which the number of individuals with a cancer diagnosis is much smaller than that without [6]. Traditional algorithms may be biased toward the dominant class, leading to suboptimal performance in cancer prediction. Additionally, cancer data frequently exhibit nonlinear relationships between the features and the outcome, but conventional algorithms such as LR and LDA assume a linear association, which can result in poor performance when applied to cancer prediction tasks. Overfitting is another concern, as traditional machine learning models are prone to capturing noise in the data rather than the underlying patterns, leading to suboptimal generalization performance and diminished predictive accuracy for cancer prediction.

Ensemble learning, which combines multiple base models to improve predictive performance, has emerged as a promising approach for building robust and accurate cancer prediction models. Ensemble methods can leverage the strengths of multiple base models, each of which can capture different subsets of relevant features, leading to a more comprehensive feature selection [7]. They can also combine the predictions of multiple models, each of which can handle imbalanced data in different ways, resulting in an improved overall performance [8]. Moreover, ensemble methods such as RF and GB can capture nonlinear relationships in the data more effectively than individual models, making them better suited for cancer prediction tasks. By combining the predictions of multiple models, ensemble methods can reduce the risk of overfitting and improve generalization performance [9]. Stacking, a popular ensemble technique, involves training a metamodel to learn the optimal way of combining the predictions of a set of base models, often outperforming individual models [10].

However, the widespread adoption of such complex machine learning models, considered “black-box” models in clinical practice, has been hindered by the lack of transparency and interpretability, which are key requirements for medical decision making [11]. Explainable AI (XAI) techniques are essential for providing insights into the influential features and decision-making process of the models, addressing the interpretability challenge, and enabling clinicians to trust and effectively utilize the predictions [12].

This study aimed to design and implement a multi-cancer prediction model using stacking with explainable AI techniques to address these challenges. To predict lung, breast, and cervical cancer, we developed and evaluated 12 machine learning models, including ensemble learning, such as RF, ET, GB, and ADB. We then propose two stacking-based models with different combinations of constituent base learners that combine the strengths of these base models to achieve higher accuracy. Importantly, we employed XAI methods such as SHAP and feature importance analysis to provide insights into the influential features and decision-making process of the stacking model.

The major contributions of this paper are as follows:The design and implementation of a comprehensive ensemble learning framework for multi-cancer prediction, including 12 base models.The rigorous evaluation of the models using various performance metrics, including accuracy, recall, precision, F1-score, AUC, Kappa, and MCC statistics.Based on the performances of the base models, two stacking-based metamodels are designed and extensively evaluated.The transparent assessment of feature importance and the impact of hyperparameters using explainable AI techniques, such as SHAP and feature importance analysis.

The remainder of this paper is structured as follows: Section 2 reviews related work. In Section 3, the proposed methodology is outlined, along with a summary of the ensemble learning approach considered in this study, the experimental setup, and the evaluation metrics used to assess the efficacy of the prediction models. Section 4 provides detailed information about the datasets of the three cancer diseases. Section 5 presents the experimental details, including the model building process and the results. It also compares the stacking model with the base models and other related papers. An extensive discussion on the interpretability of the stacking model using learning curves and SHAP is presented in Section 6. This paper concludes in Section 7, where the limitations of this study are discussed, and the potential directions for future research are identified. Table 1 contains a list of the abbreviations used throughout the paper.

## 2. Related Work

The prediction of cancer using machine learning and ensemble learning techniques has been an area of extensive research, with various approaches applied to various cancer diseases. Traditional machine learning methods have been widely used for cancer prediction, including lung cancer [13,14], breast cancer [15,16], and cervical cancer [17,18]. Many researchers have focused on ensemble learning methods, applying these techniques to image and clinical data for the improved prediction of lung [19,20], breast [21,22], and cervical [23,24] cancer.

For lung cancer prediction, Ahmad and Mayyaa [25] utilized bagging and randomized node optimization, considering risk factors such as genetic predisposition, air pollution, smoking, and other clinical symptoms. Safiyari and Javidan [26] predicted lung cancer using five ensemble methods, including bagging and ADB, applied to the SEER data. Wang et al. [27] proposed an ensemble learning model for lung cancer prognosis using RF and self-paced learning bootstrap, integrating high- and low-quality samples. Siddhartha et al. [28] employed bagging-based ensembles to predict lung cancer patients’ survival, developing a machine-learning pipeline for imbalanced data using RF. Mamun et al. [29] compared several ensemble methods, including XGB, LGBM, bagging, ADB, for lung cancer prediction, achieving the highest accuracy of 94.42% using XGB.

In the context of breast cancer, Jaiswal et al. [30] introduced an enhanced XGB ensembling (I-XGBoost) approach, incorporating big data analytics for the prediction and diagnosis of breast cancer. The study focuses on identifying the most effective features for classifying cancer as either malignant or benign. The results demonstrate that I-XGB achieves an exceptional accuracy of 99.84% when utilizing Spark’s Python API. In another study, Rabiei et al. [31] developed a model for breast cancer prediction by exploring the use of multifactorial data and optimization techniques. They used demographic, laboratory, and mammographic data comprising 5178 independent records and 24 features, thereby addressing multiple factors that may influence breast cancer diagnosis. Among all the models, RF achieved the highest accuracy rate of 80%. Although RF showed higher sensitivity (95%), its AUC (56%) indicates that the model may not generalize well across new data. Furthermore, Nemade and Fegade [20] evaluated several machine learning models to predict breast cancer using the WDBC dataset. The study compares a range of standard and ensemble machine learning models, including NB, LR, SVM, KNN, DT, RF, ADB, and XGB. Out of all the models, XGB achieved a high accuracy of 97%, with an AUC score of 99.9%, making it one of the best-performing models. However, the study lacks a detailed discussion of hyperparameter tuning and does not thoroughly examine feature engineering techniques. Naji and Filali [32] compared five machine learning models, including SVM, RF, LR, DT, and KNN, for predicting breast cancer. The study identifies SVM as the best-performing model with accuracy, precision, recall, and AUC at 97.2%, 98%, 94%, and 96.6%, respectively. Incorporating XAI techniques would be crucial for making the results more interpretable for clinical applications. Fu et al. [33] proposed an optimized version of the XGB algorithm for survival analysis, called EXSA. This model is designed to handle tied events and predict disease progression in breast cancer patients using clinical data from the CRCB at West China Hospital of Sichuan University. The models, including EXSA, Cox proportional hazards model, RSF, and GB, were compared using various evaluation metrics like the concordance index (C-index) and time-dependent AUC to assess their performance in predicting disease progression. However, the model’s complexity and the lack of focus on explainability may limit its practical application in clinical settings.

For cervical cancer prediction, Uddin et al. [23] proposed a prediction method using machine learning techniques like SVM, RF, KNN, DT, NB, LR, ADB, GB, MLP, NCC, and voting. In this work, the authors employed a hybrid feature selection strategy that includes PCA, SelectKBest, and XGB to select the most important features for predicting cervical cancer. Among all the models, voting (RF and MLP) achieved the highest accuracy of 99.19%, precision, recall, and F1-score of 100%. The work can be extended by using XAI to understand the key features that could lead to better prevention and intervention strategies for obesity. Alam et al. [34] proposed a decision-tree-based model for cervical cancer prediction. They used the BDT, DF, and DJ models and applied preprocessing to remove some attributes with high ratios of missing values and handle imbalanced classes using the SMOTE. The BDT performed well by achieving the highest accuracy rate of 94.1. The study does not compare their performance with more advanced machine/ensemble learning techniques nor addresses XAI, which is particularly important in medical applications to ensure transparency and trust in the model’s predictions. In another study, Song et al. [35] introduced an ensemble approach to cervical cancer prediction, incorporating a voting strategy among multiple classifiers and a gene-assistance module. Although the voting approach achieved the highest accuracy (83.16%) and F1-score (32.80%), the low recall might limit the model’s clinical utility, which is critical in medical diagnostics. This work can be extended using XAI to understand the most key features that could lead to better prevention and intervention strategies for cervical cancer. Furthermore, Jahan et al. [36] explored various machine learning models, including MLP, RF, KNN, DT, LR, SVC, GBC and ADB, to predict cervical cancer using risk factors and assessed their performance using accuracy, recall, precision, and F1-score. MLP outperforms other models by achieving the highest accuracy of 98.10%, precision, recall, and F1-score of 98% on the top 30 features. The work does not use any data balancing techniques to address the potential issue of class imbalance present in the dataset, which is often a common challenge in medical datasets like cervical cancer detection. Furthermore, Alsmariy et al. [37] proposed the ensemble model to predict cervical cancer based on risk factors. Key methods like SMOTE and PCA are used to handle data imbalance and dimensionality reduction, respectively. The highest accuracy of 98.49% was achieved for the Schiller test using the SMOTE-voting-PCA model. This work has limitations regarding feature importance, which is crucial to understanding how the machine learning model makes decisions, especially in medical diagnostics like cervical cancer prediction.

In recent years, machine learning and ensemble learning models combined with XAI have gained significant attention in cancer prediction research. These models can significantly enhance the understanding of key features driving predictions, providing clear- and human-understandable explanations for decisions made by models in multiple cancer prediction scenarios. In one of the recent works, Zhang et al. [38] used LR, NB, RF, and XGB to predict lung cancer and applied SHAP to visualize the contribution of each feature. Ali et al. [24] introduced a novel ensemble machine learning classifier for cervical cancer detection that integrates RF, SVM, GNB, and DT. They used the SMOTE function to improve model performance on imbalanced datasets and applied the SHAP method that strengthens the model’s interpretability and clinical relevance. However, feature selection and importance methods like RFE can be used to identify the key features and their contribution to the model’s performance. In addition, Aravena et al. [39] presented an innovative hybrid methodology combining XGB and SHAP to address the prevention of breast cancer through a machine learning-driven decision support system. The model’s primary strength lies in its explainability, which is critical for gaining acceptance in medical fields. They utilized and compared four models, including XGB, LR, RF, and SVM, using 10-fold cross-validation. It was found that XGB performed best in terms of accuracy, precision, and recall at 85%, 85.4%, and 79.5%, respectively. However, the dataset focuses on Indonesian women, which may limit the generalizability of the model to other populations with different genetic, environmental, and cultural backgrounds. Makubhai et al. [40] emphasized the use of XAI techniques, such as decision trees, PDP, and SHAP values, to provide interpretability for models such as DT, LR, RF, and XGB to predict lung cancer risk. Among the models used, XGB achieved the highest accuracy of 95.5%. This model was able to efficiently predict lung cancer risk by incorporating various patient factors like age, smoking history, and exposure to environmental toxins. While the study reports accuracy as the primary performance metric, it does not provide detailed insights into other crucial metrics such as recall, precision, F1-Score, and AUC, which are important for evaluating model performance in medical applications.

Despite the advancements in machine learning and ensemble learning for cancer prediction, several challenges remain unaddressed. First, many existing studies focus on a single type of cancer, which limits the generalizability of their findings to other cancer types. Additionally, the majority of research relies on traditional evaluation metrics such as accuracy and precision, with limited attention given to more comprehensive performance assessments like AUC, F1-score, and precision-recall trade-offs. Furthermore, the use of limited datasets, often specific to a particular population, raises concerns about the model’s applicability across different demographics and clinical conditions. Few studies have incorporated a broad range of cancer types or systematically evaluated model performance across diverse patient cohorts.

Moreover, while some recent efforts have introduced XAI techniques to improve model transparency, many studies still lack thorough explainability, which is crucial for clinical applications. The absence of feature importance analysis and insufficient use of interpretable models like SHAP hinder the practical implementation of these models in healthcare settings. Ensemble learning techniques, such as stacking, have been shown to improve predictive accuracy, but their adoption in multi-cancer prediction remains limited, especially when combined with XAI. Therefore, this study addresses these gaps by developing a comprehensive framework for predicting multiple cancer types—lung, breast, and cervical—using ensemble learning methods, including stacking, and incorporating XAI techniques to ensure model transparency and trustworthiness in clinical decision making.

## 3. Research Methodology

This section provides a thorough summary of the research methodology implemented and a brief overview of the stacking method utilized in the experiment.

### 3.1. Research Workflow

The experiment began by collecting datasets from three distinct data sources for three cancer types. After usual data preprocessing and splitting, the rest of the study was carried out in three phases:Phase I: We initially implemented twelve prediction models using different machine learning algorithms: NB [41], SVM [42], QDA [43], Ridge [44], KNN [45], LDA [46], LR [47], DT [48], GB [49], ADB [50], RF [51], ET [52]. Each model was implemented on each of the three datasets, with optimization performed through hyperparameter tuning and stratified k-fold cross-validation.Phase II: In this phase, we developed a stacking model:
(a)Initially, six randomly picked learners were used to construct the stacking model.(b)Next, the top six models, which had demonstrated the best performance on each dataset, were selected as constituent learners for the stacking model.In both scenarios, the model was independently evaluated on each dataset using a comprehensive set of performance metrics, followed by optimization via hyperparameter tuning and stratified k-fold cross-validation.

Phase III: Finally, we applied XAI techniques to interpret the outcomes of the proposed stacking model.

Figure 1 illustrates the flow of this experimental study.

### 3.2. Stacking

Stacking is a robust ensemble learning technique that significantly improves predictive accuracy by combining the strengths of multiple models. The stacking process begins by training a diverse set of base models, also known as level-0 models, on the same dataset. These base models can encompass a wide range of types, including decision trees, neural networks, and support vector machines, each providing a unique perspective on the data. After training, these models generate predictions on a hold-out set or through cross-validation, transforming these predictions into meta-features for the subsequent stage of the stacking process.

In the next stage, a higher-level model, referred to as the metamodel or level-1 model, is trained using these meta-features as inputs and the true labels as outputs. The metamodel plays a crucial role in synthesizing the predictions from the base models to produce the final output. When new data are introduced, predictions are first generated by all base models, and these outputs serve as inputs for the metamodel, which ultimately provides the final prediction. The general working of the stacking method is shown in Figure 2.

Several technical factors are crucial for optimizing the stacking process. Cross-validation, particularly *k*-fold cross-validation, is commonly employed to create out-of-fold predictions for the meta-features, ensuring that the metamodel is trained on unbiased data. The selection of the metamodel is pivotal; options can range from simple algorithms like logistic regression to more complex models such as gradient boosting. Additionally, original features from the dataset may be included alongside the meta-features, thereby enriching the input for the metamodel.

The advantages of stacking are manifold. It frequently enhances accuracy by leveraging the strengths of individual models while addressing their weaknesses. The metamodel can mitigate overfitting tendencies observed in base models, improving generalization to unseen data. Stacking also allows for the combination of diverse model types, enabling the capture of intricate patterns within the data. Furthermore, it effectively handles various data types and demonstrates robustness against the idiosyncrasies of individual models or datasets, making it a valuable tool in machine learning applications across a range of domains.

### 3.3. Experiment Setup

A computer system with Intel^®^ Core^TM^ i9-10900K CPU (3.70 GHz) was used for the experiment. The system’s other hardware specifications are-RAM: 64 GB (DDR4), HDD: 2 TB, SSD: 500GB (NVMe). The computer was running on a Windows 11 Pro operating system. Programs were written in Python (version 3.10.9) on Jupyter Notebook (version 1.0.0).

### 3.4. Evaluation Metrics

Several standard evaluation metrics, as detailed in Table 2, were used to assess the designed models’ abilities to predict three cancer types.

## 4. Dataset Description and Preprocessing

We used the lung cancer dataset provided by staceyinrobert, a publicly available dataset at Data World (https://www.kaggle.com/datasets/mysarahmadbhat/lung-cancer (accessed on 12 February 2025)). The dataset contains 309 instances and 16 demographic parameters, where the first 15 parameters are predicate variables, and the last one is the target variable. The parameters are described in Table 3, which represents the demographic, lifestyle, psychological, and clinical factors that can influence the risk and diagnosis of lung cancer. For breast cancer, we used the BCWD (https://www.kaggle.com/datasets/uciml/breast-cancer-wisconsin-data (accessed on 12 February 2025)) dataset, with 569 instances and 31 mammographic parameters. Table 4 lists the BCWD parameters that are quantitative features computed from some digitized image of a fine needle aspirate (FNA) of a breast mass. These features describe the characteristics of cell nuclei present in the image, and they are crucial for distinguishing between benign and malignant tumors. For cervical cancer, we used the Cervical Cancer (https://www.kaggle.com/datasets/ranzeet013/cervical-cancer-dataset (accessed on 12 February 2025)) dataset, which contains 835 instances and 36 related parameters. Table 5 lists the attributes of the cervical cancer dataset, which represents a mix of demographic, behavioral, and medical factors that could potentially influence the risk of developing cervical cancer.

To identify potential missing values within the dataset, we employed an imputation method, utilizing the mean for numerical values and a model-based approach for categorical values. Additionally, we applied the IQR method to detect outliers, subsequently replacing them, if present, with the mean, median, and mode values of the neighboring instances.

All three datasets were somewhat imbalanced in terms of positive and negative cancer instances. To make them balanced, we used SMOTE to upsample the minority classes, as shown in Figure 3. SMOTE was strategically selected for class balancing due to its exceptional effectiveness in enhancing minority class representation without data loss [53]. Although alternatives such as cost-sensitive learning and ensemble undersampling were considered, the proven efficacy and widespread application of SMOTE in medical datasets align well with our study’s objective of achieving practical clinical relevance [54,55].

After generating three datasets, we normalized each of them using the min–max algorithm, as stated in Equation (1). Each attribute in the datasets was assigned values between 0 and 1.(1)$$x_{scaled}=\frac{x-x_{min}}{x_{max}-x_{min}}$$
where *x* denotes the attribute value, and *x_min_* and *x_max_* represent the attribute’s minimum and maximum values, respectively.

## 5. Prediction Models and Results

In this section, we present the details of the model-building process and corresponding results. As discussed in Section 3.1, the study is divided into two phases: first, we developed twelve prediction models by applying various base learners to datasets for lung, breast, and cervical cancers. In the second phase, we constructed two stacking models using different combinations of base learners to evaluate their performance across the same datasets.

### 5.1. Base Learner Models

Initially, we developed twelve models by applying base learners to the three cancer datasets. As previously discussed, these models were optimized using *k*-fold cross-validation and hyperparameter tuning. The performance of each model across the datasets, based on the evaluation metrics, is presented in Table 6. The results show that, on the lung cancer dataset, the top-performing models were LDA, LR, and GB, with an accuracy of 93.07%, 92.60%, and 90.71%, respectively. In contrast, the least accurate models were SVM (81.17%), KNN (86.10%), and DT (87.49%). For the breast cancer dataset, the highest accuracy was achieved by QDA (96.47%), ET (96.46%), and ADB (95.71%). Similarly, on the cervical cancer dataset, ADB (96.07%), LDA (95.90%), and GB (95.56%) exhibited the best accuracy. When considering the average accuracy across all three datasets, as illustrated in Figure 4, LDA was the top performer with an average accuracy of 94.64%, followed by GB (93.99%) and LR (93.92%). The performance of the other models, based on different evaluation metrics, can be interpreted from Figure 4.

### 5.2. Stacking Models

To build the stacking model, we used the base learners discussed in the previous section. For this, we adopted two approaches:In the first approach, six algorithms (SVM, KNN, DT, ET, GB, ADB) among the twelve were randomly considered. To ensure a comprehensive and robust analysis, we employed three categories of models: base models, Bagging, and Boosting on all three datasets. We selected two models from each category using the ‘random()’ function. Through careful experimentation with various permutations and combinations by executing the `random()` function repetitively, we identified the optimal combination: SVM, KNN, DT, ET, GB, and ADB.In the second, the six top-performing algorithms (based on their accuracy) were considered to build the stack. Table 7 shows the top six algorithms for each dataset, as studied in Section 5.1.

For the level-1 (meta) model in the stacking framework, we used the SVM model for all diseases using the first approach (random base models). For the second approach (best base models), we used the LR model for lung cancer, and Ridge for breast and cervical cancer, respectively. This choice was made due to its simplicity, interpretability, and strong generalization performance, especially when combining the outputs of diverse base learners. These meta-learners also tend to work well with the meta-features derived from the probabilistic outputs of the base models.

In both cases, to remove the bias in the datasets, we employed stratified *k*-fold cross-validation. The procedure of *k*-fold cross-validation is depicted in Figure 5. The optimal value of *k* was found to be 10 for all cases.

To optimize model performance, we employed two established hyperparameter tuning techniques: grid search and random search. Grid search exhaustively evaluates all possible combinations within a predefined hyperparameter space, ensuring systematic exploration of the parameter landscape. In contrast, random search samples a fixed number of random combinations from the same space, offering computational efficiency at the cost of reduced thoroughness.

For grid search, we constructed a parameter grid tailored to the nature and range of its hyperparameters for each algorithm. For instance, in the case of the NB classifier, eight key hyperparameters were considered, including random_state (binary), *c* (integer), gamma (decimal), kernel (linear, polynomial, RBF), probability (binary), verbose (binary), refit (binary), and verbose levels (5 discrete levels). To maintain computational feasibility, we limited the range of integer and decimal parameters to a maximum of 12 values each. This resulted in a comprehensive search space, such as 2 × 12 × 12 × 3 × 2 × 2 × 2 × 5 combinations for the NB model. This exhaustive pattern of evaluation was consistently applied across all other models included in the study, ensuring that possible interactions between hyperparameters were systematically explored.

In contrast, random search samples hyperparameter combinations stochastically from the same predefined parameter sets. In our experiments, we generated 12 random combinations for each model, allowing us to compare the efficiency and effectiveness of this approach with grid search. While a random search offers greater computational efficiency and can be advantageous in high-dimensional spaces, its stochastic nature may overlook the critical regions of the hyperparameter space that grid search is able to systematically evaluate.

Our experiments demonstrated the superiority of grid search, which consistently produced better-optimized models. This performance advantage likely stems from two key factors. First, grid search’s exhaustive nature enables the precise identification of optimal hyperparameter interactions—particularly critical in ensemble methods, where parameters such as the learning rate and estimator count exhibit strong interdependence. Second, unlike random search, grid search guarantees that no high-potential configurations are missed due to sampling bias.

The final set of optimal hyperparameters for each ensemble model, as determined through grid search, is presented in Table 8.

We assessed the feature importance for the three datasets using RFE. It was found that most of the features were significant for the lung cancer dataset. For the breast cancer dataset, nearly 15 features among 31 were significant, and the rest had lesser contribution. Therefore, no features were eliminated from these two datasets. However, in the cervical cancer dataset, 7 features among 33 had zero contribution. We eliminated the non-contributing features from the cervical cancer dataset, resulting in 26 features remaining to work with.

The confusion matrix for the stacking models using both approaches is shown in Table 9. Figure 6 shows the average performance (of ten folds) of the stacking models using both approaches for each dataset. Overall, both models performed better with the cervical cancer dataset. However, both models achieved 100% precision with the lung cancer dataset, but the recall scores are the lowest with this dataset. Figure 7 shows the performance deviations of the stacking models across ten folds for each dataset. In this case also, both the models performed better with the cervical cancer dataset. A lower standard deviation value implies that the model’s prediction performance is consistent across the folds.

Next, we compared the two stacking models. Figure 8 shows the average performance of both models for all three datasets. It is clearly depicted that the stacking model with top six base learners outperformed the stacking model with the randomly selected six base learners in all parameters. The stacking model with top six base learners is also far more consistent across the folds than the stacking model with randomly selected six base learners, as shown in Figure 9. Therefore, for further analysis, in our study, we considered the stacking model with second approach (henceforth referred only as stacking model), i.e., when the top-performing base learners are used to build the stacking model.

The AUC-ROC curves of the proposed stacking model for lung, breast, and cervical cancer datasets are shown in Figure 10, Figure 11 and Figure 12, respectively. Overall, the stacking model performed equally well across all datasets. However, it achieves the best result for the breast cancer dataset.

### 5.3. Comparative Analysis

This section provides a comprehensive comparison of the proposed hybrid voting and stacking models with other designed models in this study, as well as state-of-the-art literature.

#### 5.3.1. Comparing with Other Models

This section compares the stacking models proposed in Phase II with the base learners considered in Phase I. To keep the comparison precise, we selected the top three performing base learners for each metric, separately for each dataset. For example, LDA, LR, and GB had the best accuracies on the lung cancer dataset, whereas QDA, ET and ADB had the best accuracies on the breast cancer dataset, and on the cervical dataset, LDA, ADB and GB had the best accuracies.

The comparative performances in terms of accuracy, recall, precision, F1-score, AUC, Kappa, and MCC are shown in Figure 13, Figure 14, Figure 15, Figure 16, Figure 17, Figure 18 and Figure 19. Both the stacking models by far outperformed the base learners on all the metrics for all three cancer datasets. However, for the lung cancer dataset, the stacking models exhibited lower recall values than the base learners. This might be due to their optimization for higher precision (100%). Furthermore, the stacking model with the top six base learners performed better than the stacking model with the randomly selected six base learners. The average performance comparison of the proposed stacking models with the top three base learners for all three datasets is shown in Figure 20.

#### 5.3.2. Comparing with State-of-the-Art

Table 10 presents the performance comparison of our proposed stacking model with similar research papers in cancer prediction (lung, breast, and cervical). For comparison, only those papers are considered that have applied at least one ensemble model in lung, breast, or cervical cancer prediction using demographic or lifestyle data.

Our stacking model achieved an impressive 99.94% accuracy, surpassing the best-performing models from other studies, such as Wang et al. [27] (97.96%), Uddin et al. [6] (99.16%), and Ali et al. [11] (98.06%). Furthermore, our model achieved 100% precision, 99.47% recall, and 99.73% F1-score, marking its superior performance in identifying true positive cases while minimizing false positives. The AUC score of 99.74% also highlights its excellent discriminatory power. In contrast, models such as those by Patra [14] and Siddhartha et al. [28] displayed a much lower performance, with accuracy around 81.25% and 84.04%, respectively, underscoring the robustness of our approach.

Our paper stands out due to the comprehensive inclusion of a wide variety of best-performing base learners (e.g., LR, CB, XGB, RF, ET) to build the stacking model. Our model’s enhanced performance with stacking can also be attributed to a well-optimized meta-learner, efficient cross-validation, and fine-tuned hyperparameters.

## 6. Model Interpretation

Building on the findings from the previous section, it is essential to examine the impact of clinical and demographic factors on the predictive capabilities of the prediction model in cancer risk assessment. In this section, we evaluate the effectiveness of the proposed stacking model by analyzing learning curves and utilizing XAI tools. These tools shed light on the models’ performance trends and provide valuable insights into how specific features contribute to the predictions, deepening our understanding of the roles that individual predictors play in shaping overall model outcomes.

### 6.1. Learning Curves

The learning curves for the stacking model applied to the lung, breast, and cervical cancer datasets are depicted in Figure 21, Figure 22 and Figure 23, respectively. These curves illustrate the model’s performance as more data or iterations are introduced, helping assess whether the model is overfitting or underfitting. The training and validation curves provide a clear understanding of the reliability of the stacking model’s learning process.

The figures reveal that the training and validation curves for the stacking model follow smooth, nearly linear trajectories, suggesting stable learning behavior. However, for the breast cancer dataset, the validation curve dips slightly toward the end of the data points, while the training curve remains completely flat. This flatness in the training curve indicates potential overfitting, as the model may be fitting too closely to the training data without generalizing well to unseen data. In contrast, for the cervical cancer dataset, both the training and validation curves converge, which is an encouraging sign of minimal overfitting or underfitting. This convergence indicates that the model is learning effectively and generalizing well to new data.

### 6.2. XAI

XAI refers to a set of methods and techniques that aim to make the decisions and processes of AI models more transparent, interpretable, and understandable to human experts. Using XAI, the prediction models can be interpreted at the global and local levels. Both local explanations and global explanations are essential to ensure that models are transparent, trustworthy, and practically useful. Each type of explanation serves a different purpose, addressing distinct needs within the medical and healthcare domains. A global explanation can highlight feature importance on a large scale, helping clinicians and researchers understand which factors are generally influential in predicting disease outcomes. Local explanations can then dive deeper into how those same factors affect individual patients, bridging the gap between research and clinical practice.

In this work, we used the SHAP method to derive the explainability and interpretability of the proposed stacking model for lung, breast, and cervical cancer prediction. SHAP is a method used to explain the interpretability of AI-based models, such as those based on machine learning, deep learning, transfer learning, and others. It primarily aims to explain individual predictions, like disease predictions in our case, by leveraging the game’s theoretically optimal Shapley values. These values, originating from cooperative game theory, are popular due to their favorable properties. In this context, the feature values of a data instance are considered as players in a coalition, and the Shapley value represents the average marginal contribution of a feature across all potential coalitions.

#### 6.2.1. Global Explanation

A global explanation offers valuable insights into the overall behavior of an AI model across the entire patient population. It focuses on the general patterns and the relationships between the features (e.g., age, genetic markers, lab results) and the model’s predictions throughout the dataset. This type of analysis reveals which features are most significant in the model’s decision-making process, ensuring alignment with established medical knowledge. Such an overarching understanding helps validate the model’s predictions and identify any discrepancies that may need further tuning.

Global explanations also play a critical role in detecting bias, revealing whether the model unfairly impacts specific demographic groups, which allows for adjustments to promote fairer outcomes. Additionally, they are key to meeting regulatory and ethical standards, ensuring that the model’s behavior complies with legal frameworks and healthcare regulations such as GDPR, HIPAA, or FDA requirements. By enhancing transparency and consistency in medical decision making, they help build trust in AI models.

For global explanation, we utilized the mean absolute SHAP feature importance that ranks the features based on their overall contribution to the predictions across instances. It measures how strongly each feature influences the predictions, regardless of whether it pushes them toward the positive or negative class. Shapley values from cooperative game theory are used in SHAP to explain the individual predictions of black box models by calculating the average contribution of each feature. By taking the absolute value, we focus on the strength of the influence without considering the direction, making it easier to compare the relative importance of features. The SHAP values are calculated using Equation (2) [57].(2)$$\varphi_{i}\left(f,x\right)=\sum_{s\subseteq F\backslash \{i\}}\frac{\left|s\right|!\left(|F|-\left|s\right|-1\right)!}{F!}{\;[f}_{s\cup i}\left(x_{s\cup i}\right)-f_{s}\left(x_{s}\right)]$$
where $\varphi_{i}$ is the SHAP value for feature *i*, *f* is the blackbox model, *x* is the input datapoint, *F* is the set of all features, *s* is a subset of features not containing the feature *i*, $f_{s}\left(x_{s}\right)$ is the model trained with features in subset *s* and evaluated on the values of those features from input *x*,${\;f}_{s\cup i}\left(x_{s\cup i}\right)$ is the model’s output when we add feature *i* to subset s, and *F\{i}* is all the possible feature subsets not containing *i*.

Figure 24, Figure 25 and Figure 26 illustrate the importance of features across the lung, breast, and cervical cancer datasets. The features are arranged in descending order of importance along the *y* axis, while the *x* axis reflects the mean SHAP values, allowing us to assess the impact without regarding the direction.

In the lung cancer dataset (Figure 24), it is evident that all features influence the risk of developing the disease. Notably, *fatigue* has the highest mean SHAP value of +1.52, indicating it is the most influential feature in predicting lung cancer risk. *Alcohol consumption*, with a value of +1.35, is a close second, significantly impacting risk assessment. *Allergy*, at +1.11, is also critical, though slightly less impactful than *fatigue* and *alcohol consuming*. Conversely, features like *chest pain*, with a value of +0.21, has minimal effect on the prediction.

Figure 25 shows that *worst concave points*, *mean concave points*, and worst perimeter are the most contributing features for breast cancer, with mean SHAP values of +2.79, +2.34, and +1.89, respectively. In contrast, *smoothness error*, *mean compactness*, *worst fractal dimension* and *mean perimeter* have a mean SHAP value of +0.01, indicating their negligible effect on breast cancer. In Figure 26, regarding cervical cancer, the *Schiller test* stands out with a mean SHAP value of +4.15, playing a pivotal role in predicting disease risk. Other features like *age* and *first sexual intercourse* also contribute positively, though their influence is less pronounced. Many features exhibit negligible impact, with values close to zero, such as *condylomatosis* (+0.01), *IUD* (+0.01), *STD* (+0.01), *HIV* (+0.01), and *syphilis* (+0.01).

#### 6.2.2. Local Explanation

A local explanation provides insights into the prediction of a specific instance or case (e.g., a particular patient’s diagnosis). This is crucial in healthcare because every patient is unique, and medical decisions are often personalized. Local explanations help doctors make decisions for individual patients. It helps answer the question: “Why did the model predict this specific outcome for this patient?”

Local explanations enable clinicians to understand the specific factors, such as biomarkers or medical history, that contributed to a diagnosis, allowing for more tailored treatment plans. By offering case-specific reasoning, local explanations enhance trust and transparency between doctors and patients, particularly in critical medical decisions. They also help identify and mitigate errors, revealing which features may have led to misclassification and allowing for corrective interventions. Furthermore, local explanations can be cross-validated against clinical knowledge, helping medical professionals ensure that the model’s predictions align with established research, thus reducing the “black-box” nature of AI and promoting its use in diagnostic processes. This level of interpretability is essential in medical domains, where trust in model predictions is vital for clinical decision making.

In this study, we employed SHAP waterfall and force plots to provide local explanations for the designed stacking model used in lung, breast, and cervical cancer prediction. These plots are instrumental in interpreting individual predictions by showing how each feature contributes to the final decision. The SHAP waterfall plot offers a clear, step-by-step breakdown of the cumulative effect of each feature on the prediction, starting from a base value (the average model output) and moving towards the final predicted value. The SHAP force plot, on the other hand, provides a more visual and intuitive representation of how different features push the prediction higher or lower, making it easy to grasp which factors are driving the outcome. Both plots are essential for enhancing model transparency, enabling clinicians to interpret the importance and impact of individual features at a granular level, making them highly effective tools in personalized medicine and clinical decision making.

##### Using Waterfall Plot

The SHAP waterfall plot is a powerful tool for visualizing how individual features contribute to a model’s prediction in a highly interpretable and transparent manner. In the context of cancer prediction, this tool is particularly valuable, as it provides insights into the model’s decision making process on a case-by-case basis, enabling the identification of critical features—such as specific medical conditions or demographic factors—that significantly influence the outcome for individual patients.

In Figure 27, the SHAP waterfall plot illustrates the contributions of various features to the prediction of lung cancer. The *x* axis represents the target variable, lung cancer, while the chosen observation, *x*, has a predicted value of f(x) = −5.988. The expected value of the target variable, E[f(x)] = −3.324, reflects the average of all predictions made by the model. Among all the features, *swallowing difficulty* had the most substantial positive impact, contributing a SHAP value of +2.34, while *shortness of breath* had the least influence, with an SHAP value of −0.05. Notably, the sum of all SHAP values equals the difference between E[f(x)] and f(x), demonstrating the model’s comprehensive feature contribution.

Similarly, Figure 28 shows the SHAP waterfall plot for breast cancer prediction, where the *concave points* of the breasts emerged as the most impactful feature with a SHAP value of +1.53, while the *perimeter* and *fractal dimension* contributed no impact at all. In this instance, the model’s predicted value (f(x)) is +11.118, and the expected value (E[f(x)]) of the target variable is +4.693.

For cervical cancer prediction, Figure 29 reveals that the feature *hormonal contraceptives* had the highest impact on the model’s prediction, with an SHAP value of +0.15 for the patient in question, while features like *HIV* and *vulvo-perineal condylomatosis* had no effect on the prediction, both contributing SHAP values of −0.0.

##### Using Force Plot

While the waterfall plot emphasizes the sequential contribution of features, the force plot focuses on the overall push-pull effect of features around the base value. Figure 30, Figure 31 and Figure 32 present the SHAP force plots for lung, breast, and cervical cancers, respectively, for some specific instances. A force plot visualizes the contribution of each feature to a model’s prediction for a particular observation. Each plot represents how different features push the prediction higher or lower from a baseline value. The baseline prediction value for lung cancer (Figure 30) is approximately −3.324. This serves as the reference point for the model’s prediction. The final prediction value is −5.99, indicating a negative prediction for lung cancer risk, suggesting that this particular case is less likely to be classified as having lung cancer. Features like *yellow fingers* (2), *anxiety* (2), and *alcohol consumption* (2) are the positive contributors (marked in red), i.e., they push the prediction toward a higher lung cancer risk. These features have strong associations with lung cancer in general. The length of the arrows indicates the strength of each feature’s contribution. The value in the brackets denotes the patient’s feature values. Features such as *fatigue* (1) and *age* (63) are the negative contributors (marked in blue), pushing the prediction lower, indicating a lower risk of lung cancer. However, they are outbalanced by the stronger negative contributions of other features.

In Figure 31, the baseline value is approximately +4.693, suggesting that the model starts with a positive risk assessment, and the prediction outcome is 11.19, indicating a strong prediction for breast cancer risk. Features such as *mean texture* (20.71), *worst area* (1121), and *standard error of area* (44.41) push the prediction higher, significantly increasing the risk of breast cancer. Other features like *worst concave points* (0.1583) and *worst radius* (19.28) also contribute positively but to a lesser extent compared to *mean texture* and *worst area*.

For cervical cancer (Figure 32), the baseline prediction is approximately +5.591. The final prediction value is −6.13, indicating a negative prediction for cervical cancer risk. Features like the *Schiller test* (0), *no. of pregnancies* (7), *age* (34), and *first sexual intercourse* (15) significantly push the prediction lower, reflecting factors that contribute to a reduced risk of cervical cancer for the specific patient. There are no significant features pushing the prediction toward higher risk.

## 7. Conclusions, Limitations, and Further Scope

In this study, we developed a stacking-based multi-cancer prediction model that significantly outperformed traditional base models in terms of both accuracy and interpretability, demonstrating its practical utility in predicting lung, breast, and cervical cancers. By combining a diverse range of base learners and integrating explainable AI (XAI) techniques, such as SHAP, the model offers valuable insights into the most influential features contributing to cancer risk assessment. These insights provide both predictive accuracy and transparency, which are crucial for clinical applications. The model’s average accuracy of 99.28% for three datasets, alongside its strong precision, recall, F1-score, and AUC, highlights its robustness and reliability in distinguishing between cancerous and non-cancerous cases.

The research emphasizes the importance of a select group of highly predictive features that drive the model’s decision-making process, largely derived from demographic and clinical data. These key features offer a focused, data-efficient approach to risk assessment, potentially improving early detection and personalized treatment strategies. The use of SHAP also allowed for better interpretability, giving healthcare professionals a clear understanding of how each feature contributes to the model’s predictions. For example, clinical indicators such as worst concave points for breast cancer and Schiller test for cervical cancer were found to be highly influential, suggesting their importance in refining cancer risk models.

The results of this study can have significant clinical implications. By highlighting the most important factors contributing to cancer predictions, the model allows for the development of more streamlined, efficient screening processes. These findings have the potential to influence clinical practices by enabling more targeted interventions and improving early diagnosis rates, which are critical for effective treatment. In cancer prediction, where both accuracy and early detection can drastically affect patient outcomes, our model presents a substantial advancement over previous approaches. Additionally, its interpretability fosters trust and transparency, which are essential for integrating AI models into clinical workflows.

Despite the promising results, the study has several limitations that must be acknowledged. First, the generalizability of the findings may be limited due to the demographic homogeneity of the dataset used. Since the dataset primarily focuses on specific population groups, the model’s predictive ability across more diverse populations may be constrained. This could lead to disparities in cancer prediction when applied to different demographic groups, potentially limiting its effectiveness in a global clinical setting. Additionally, while the study focused on three major types of cancer (lung, breast, and cervical), it does not cover the wide variety of other cancer types, each of which may have different risk factors and require unique modelling approaches. The limited range of cancers analyzed restricts the broader applicability of the findings to other malignancies. Another notable limitation is the absence of certain critical lifestyle, genetic, and environmental factors that could play significant roles in cancer risk but were not included in the dataset. Incorporating a more comprehensive set of features may enhance the model’s performance and provide a more complete picture of the cancer risk assessment.

Moreover, although the study leveraged SHAP to provide insights into feature importance, the interpretability of the ensemble model itself was not fully explored. While SHAP reveals which features are most influential, it does not explain the internal workings of the stacking mechanism or the relationships between base models and their combined outputs. Lastly, potential biases in data collection, such as the underrepresentation of certain subgroups and initial feature selection, were not thoroughly addressed. These biases could affect the model’s fairness and accuracy, particularly when applied in real-world clinical settings.

Future research should aim to overcome these limitations by leveraging the opportunities that exist to build upon the findings of this research. Expanding the model to include additional types of cancers and integrating data from more diverse demographic groups could enhance its generalizability and clinical relevance. For instance, studying cancers with different etiologies—such as prostate, ovarian, or colorectal cancers—may uncover new predictive features and extend the model’s utility in oncology. Future research could also incorporate a broader range of data, including genetic, environmental, and lifestyle factors, which would allow for a more comprehensive risk assessment. The temporal stability of the identified features should be examined to ensure that the model remains relevant over time as new clinical and lifestyle data emerge. Investigating the longitudinal impact of key features on cancer progression would further refine prediction models and improve early intervention strategies. Moreover, exploring hybrid models that integrate ensemble methods with deep learning techniques could lead to enhanced prediction accuracy by capturing more complex patterns in the data. These approaches could also facilitate personalized risk assessments, tailoring predictions to individual patients based on a wider array of factors, thus improving targeted interventions and treatment outcomes.

Real-world clinical validation is essential to ensure the model’s practical applicability. Conducting prospective studies in hospital settings, where the model is tested on real-time patient data, would be a crucial next step. This would not only validate the model’s performance but also help identify any operational challenges that may arise when integrating AI-driven models into healthcare systems. Additionally, developing a user-friendly decision-support tool, incorporating the stacking model, would make it easier for healthcare professionals to use in routine clinical practice. Such a tool could assist in early cancer detection, improving treatment decisions and ultimately leading to better patient outcomes. Lastly, ethical considerations surrounding the use of AI in healthcare need further exploration. Addressing potential biases, ensuring equitable healthcare outcomes, and maintaining transparency in decision-making are all critical areas for future research. A comprehensive investigation into the societal impacts of AI-driven cancer prediction models would ensure that such technologies are implemented responsibly, promoting fairness and accessibility in healthcare.

## Figures and Tables

**Figure 1 bioengineering-12-00472-f001:**
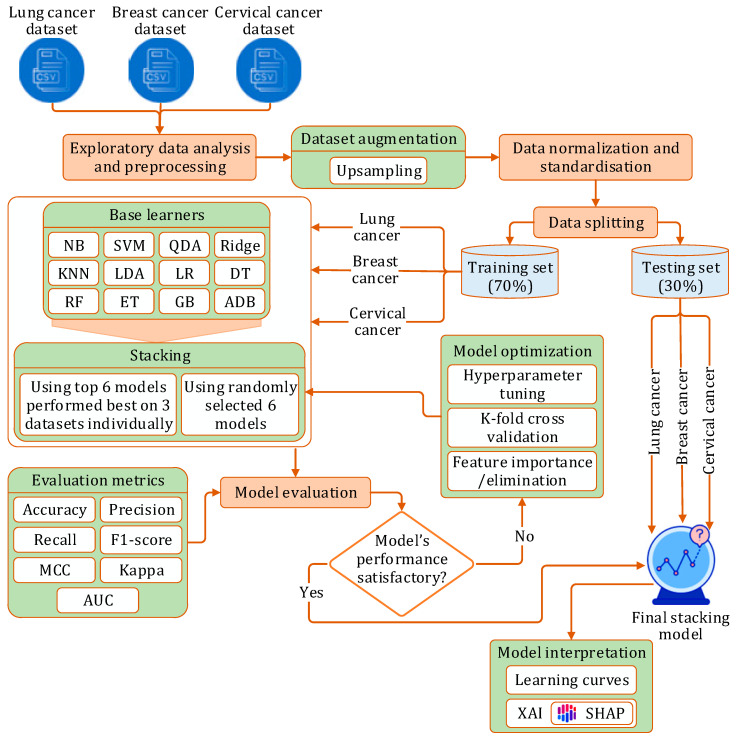
Research workflow.

**Figure 2 bioengineering-12-00472-f002:**
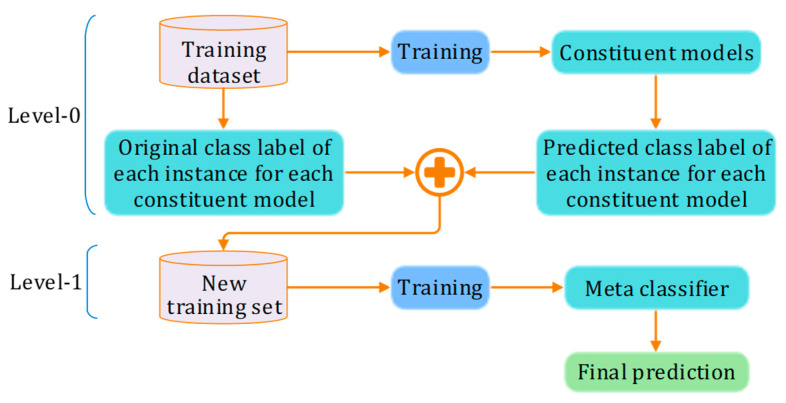
Working principle of stacking method.

**Figure 3 bioengineering-12-00472-f003:**
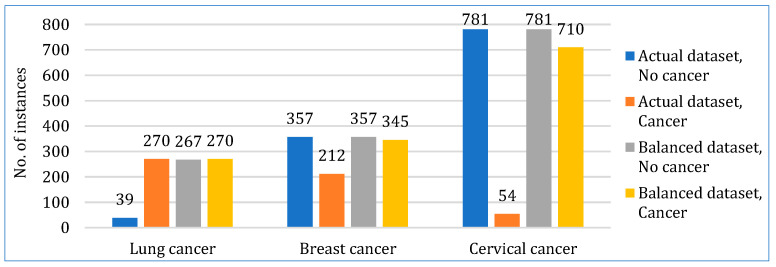
Data balancing.

**Figure 4 bioengineering-12-00472-f004:**
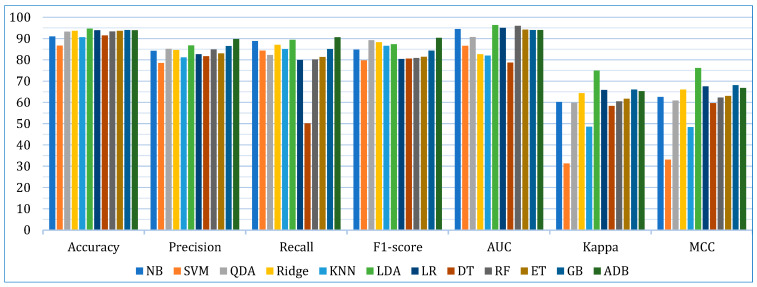
Average performance evaluation of the base models for three cancer datasets.

**Figure 5 bioengineering-12-00472-f005:**
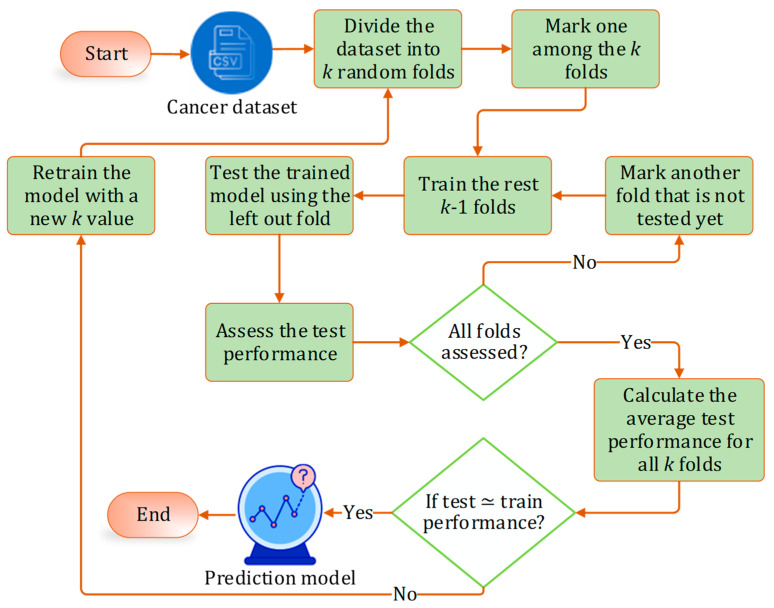
Stratified k-fold cross validation process.

**Figure 6 bioengineering-12-00472-f006:**
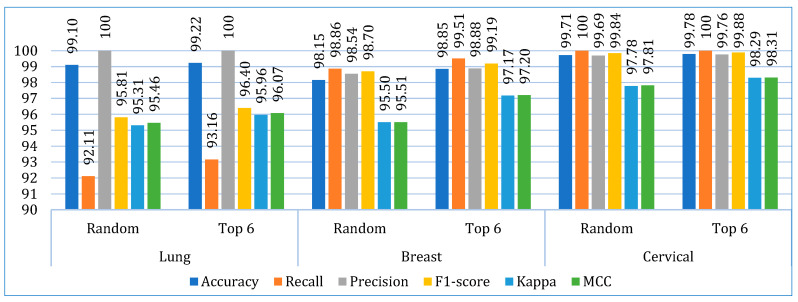
The mean performance of the stacking models (using random and top six base learners) across ten folds.

**Figure 7 bioengineering-12-00472-f007:**
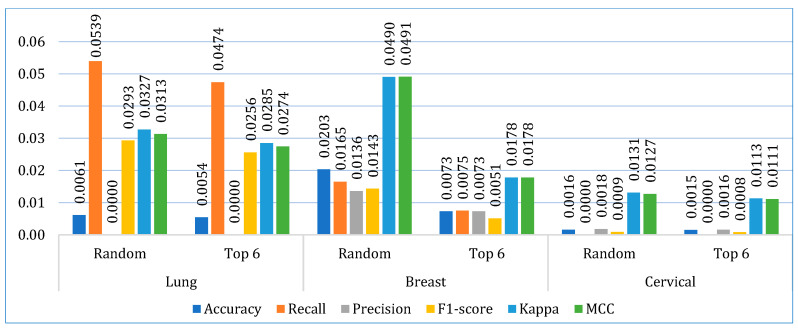
The standard deviation of the stacking models (using random and top six base learners) across ten folds.

**Figure 8 bioengineering-12-00472-f008:**
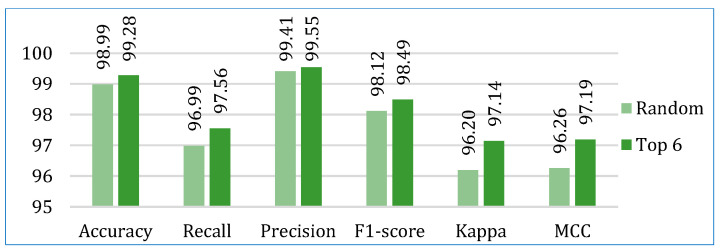
Comparing the average performance of the two stacking models (using random and top six base learners) for three cancer datasets.

**Figure 9 bioengineering-12-00472-f009:**
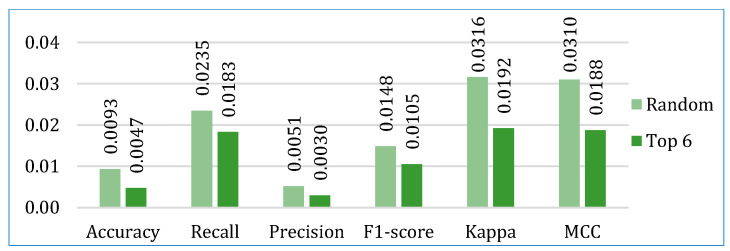
Comparing the average standard deviation of the two stacking models (using random and top six base learners) across ten folds for three cancer datasets.

**Figure 10 bioengineering-12-00472-f010:**
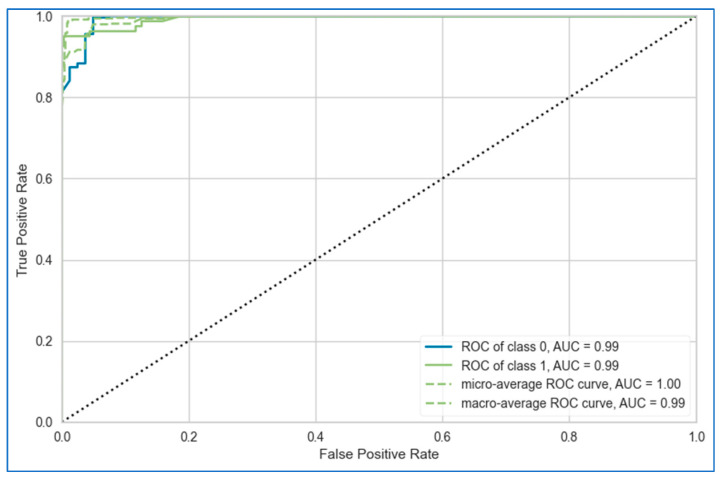
AUC-ROC curve of the proposed stacking model using top six base learners for lung cancer dataset.

**Figure 11 bioengineering-12-00472-f011:**
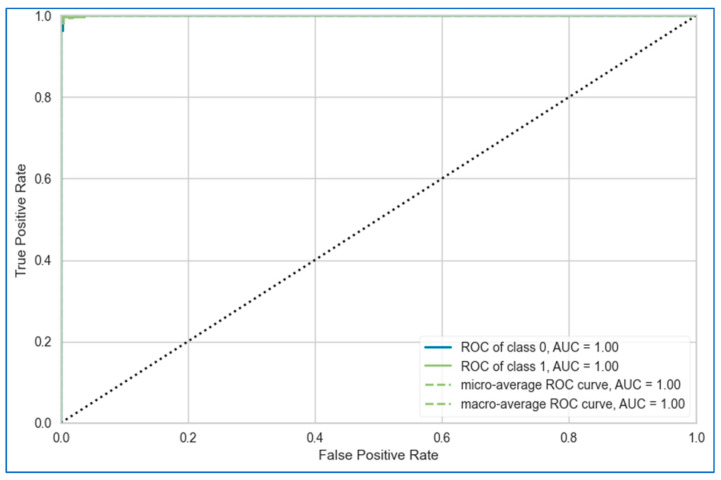
AUC-ROC curve of proposed stacking model using top six base learners for breast cancer dataset.

**Figure 12 bioengineering-12-00472-f012:**
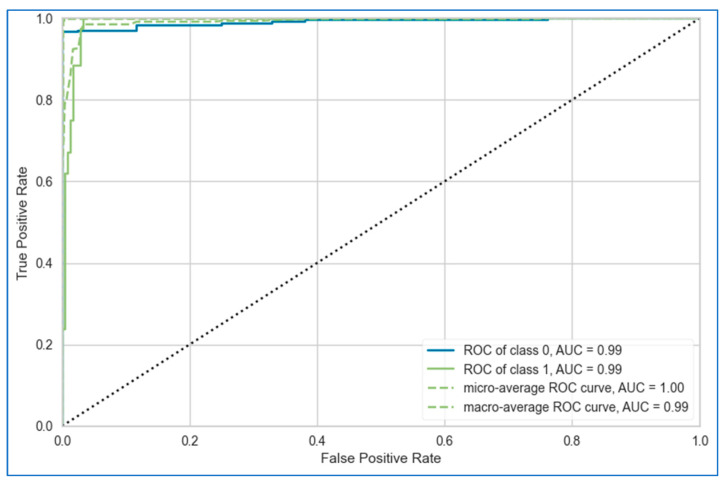
AUC-ROC curves of proposed stacking model using top six base learners for cervical cancer dataset.

**Figure 13 bioengineering-12-00472-f013:**
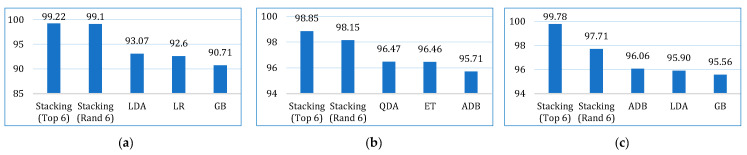
Accuracy comparison of the proposed stacking models (using both approaches) with the top three base learners for (**a**) lung dataset, (**b**) breast cancer dataset, and (**c**) cervical cancer dataset.

**Figure 14 bioengineering-12-00472-f014:**
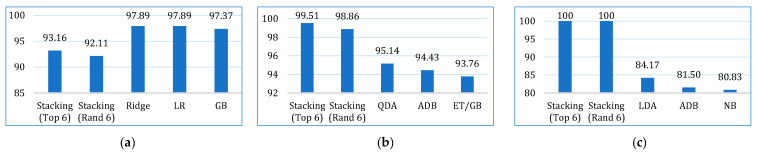
Recall comparison of the proposed stacking models (using both approaches) with the top three base learners, for (**a**) lung dataset, (**b**) breast cancer dataset, and (**c**) cervical cancer dataset.

**Figure 15 bioengineering-12-00472-f015:**
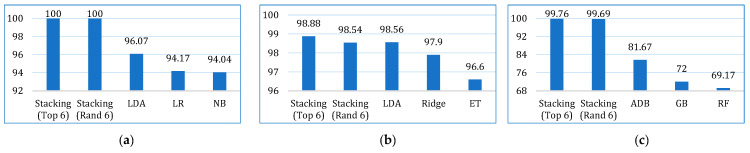
Precision comparison of the proposed stacking models (using both approaches) with the top three base learners, for (**a**) lung dataset, (**b**) breast cancer dataset, and (**c**) cervical cancer dataset.

**Figure 16 bioengineering-12-00472-f016:**
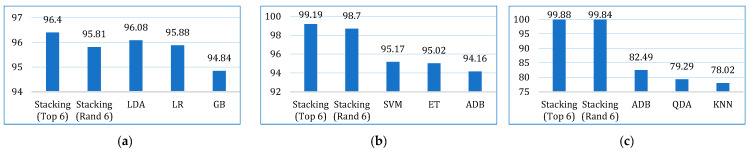
F1-score comparison of the proposed stacking models (using both approaches) with the top three base learners, for (**a**) lung dataset, (**b**) breast cancer dataset, and (**c**) cervical cancer dataset.

**Figure 17 bioengineering-12-00472-f017:**
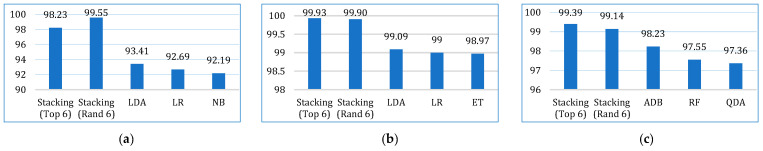
AUC comparison of the proposed stacking models (using both approaches) with the top three base learners, for (**a**) lung dataset, (**b**) breast cancer dataset, and (**c**) cervical cancer dataset.

**Figure 18 bioengineering-12-00472-f018:**
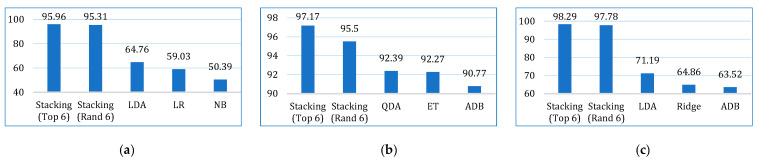
Kappa comparison of the proposed stacking models (using both approaches) with the top three base learners, for (**a**) lung dataset, (**b**) breast cancer dataset, and (**c**) cervical cancer dataset.

**Figure 19 bioengineering-12-00472-f019:**
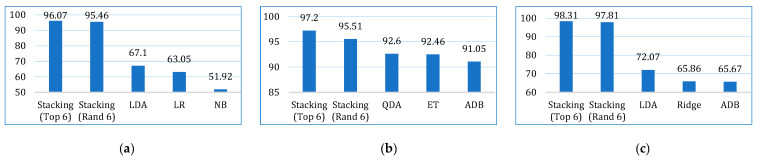
MCC comparison of the proposed stacking models (using both approaches) with the top three base learners, for (**a**) lung dataset, (**b**) breast cancer dataset, and (**c**) cervical cancer dataset.

**Figure 20 bioengineering-12-00472-f020:**
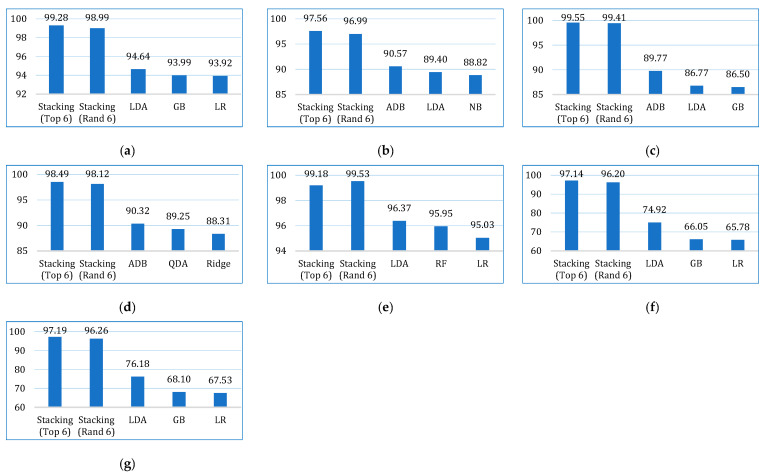
Average performance comparison of the proposed stacking models (using both approaches) with the top three base learners, for all three datasets: (**a**) accuracy, (**b**) recall, (**c**) precision, (**d**) F1-score, (**e**) AUC, (**f**) Kappa, (**g**) MCC.

**Figure 21 bioengineering-12-00472-f021:**
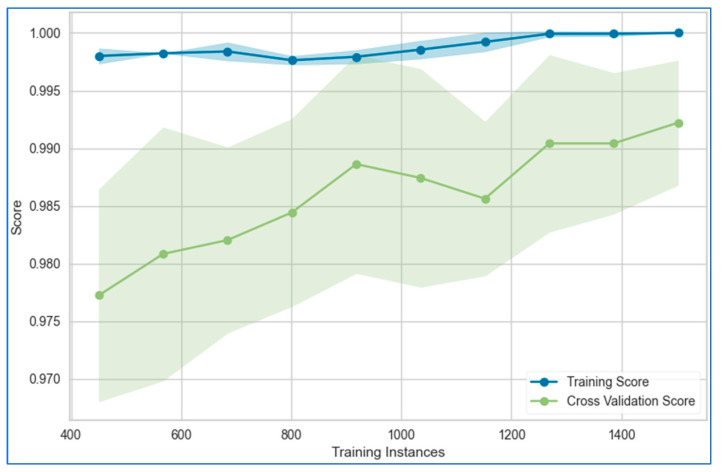
Learning curve of the stacking model using top six models for lung cancer dataset.

**Figure 22 bioengineering-12-00472-f022:**
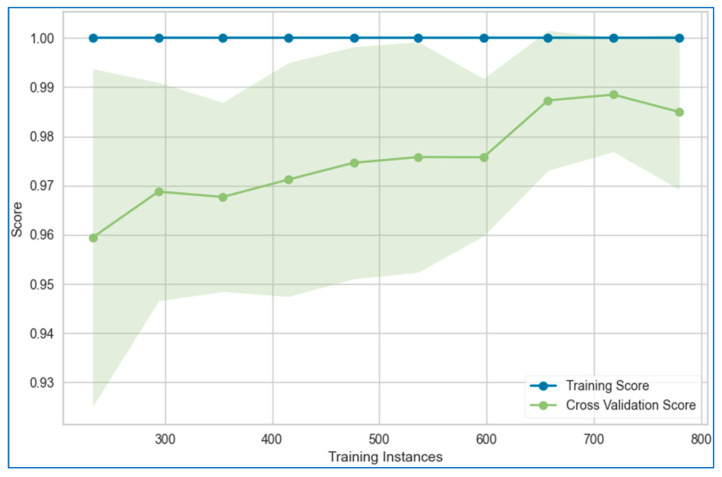
Learning curve of the stacking model using top six models for breast cancer dataset.

**Figure 23 bioengineering-12-00472-f023:**
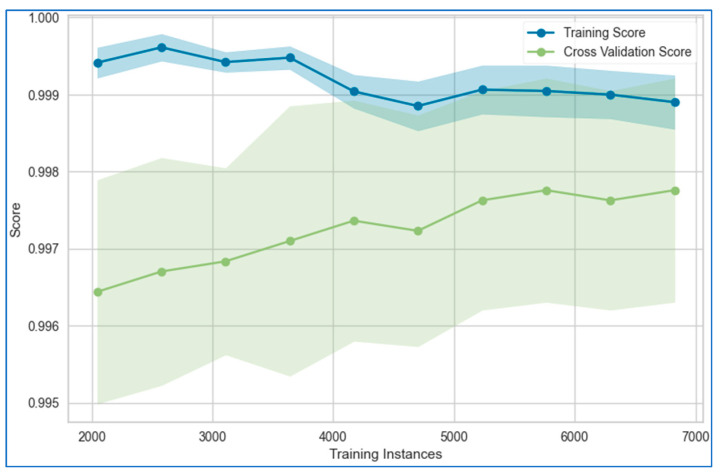
Learning curve of the stacking model using top six models for cervical cancer dataset.

**Figure 24 bioengineering-12-00472-f024:**
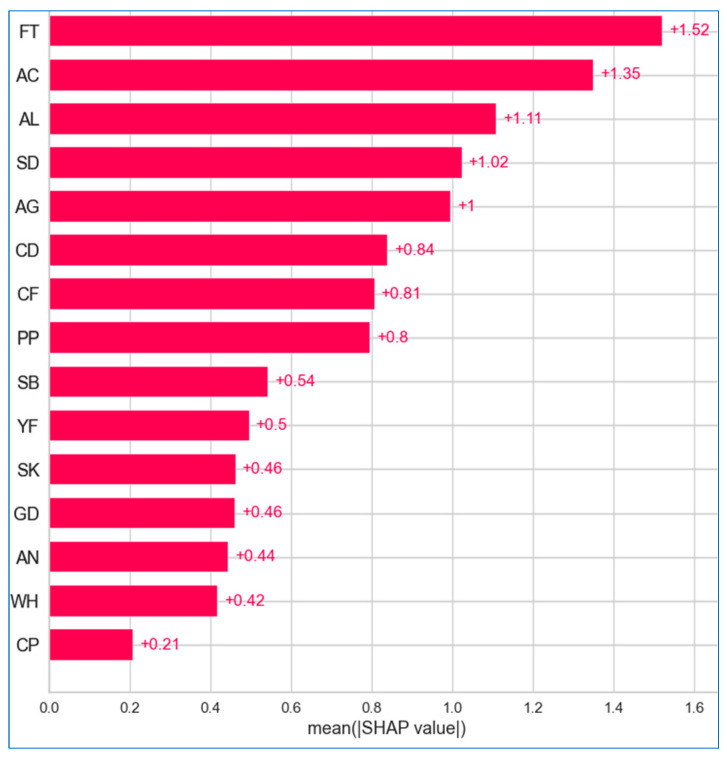
Absolute mean SHAP for lung cancer.

**Figure 25 bioengineering-12-00472-f025:**
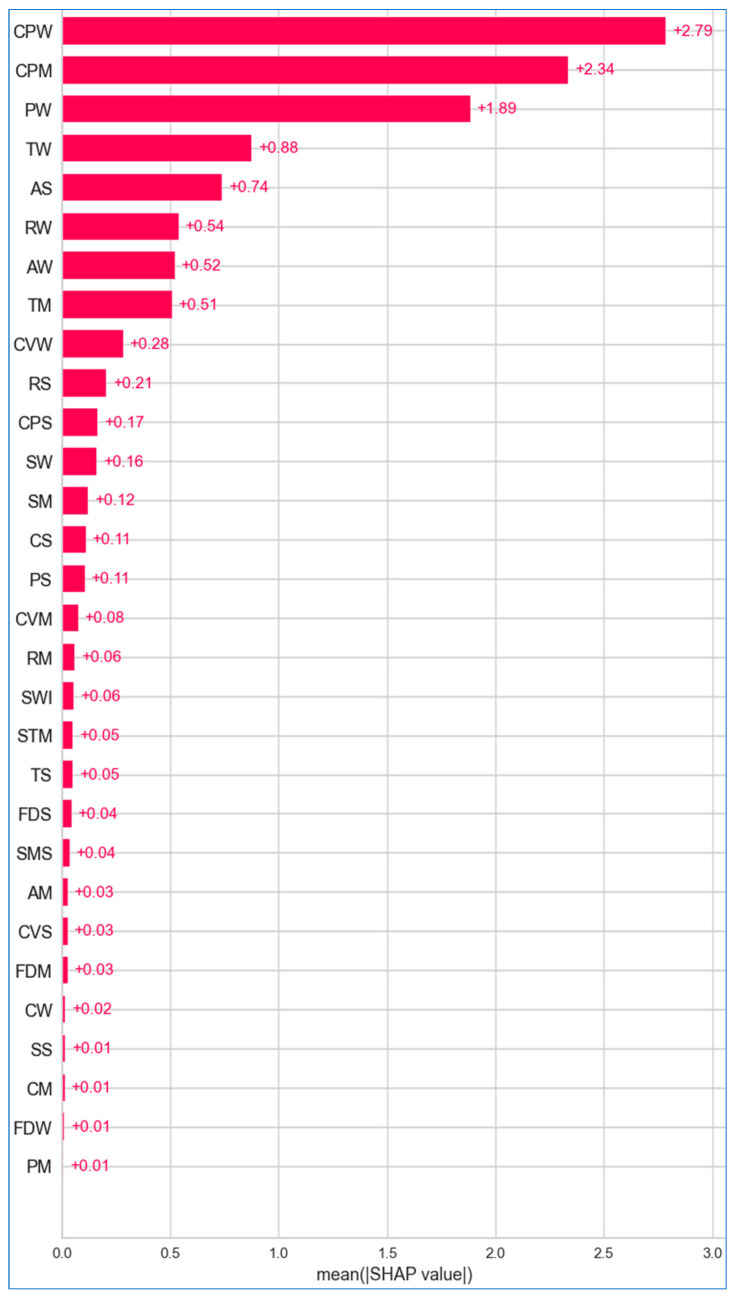
Absolute mean SHAP for breast cancer.

**Figure 26 bioengineering-12-00472-f026:**
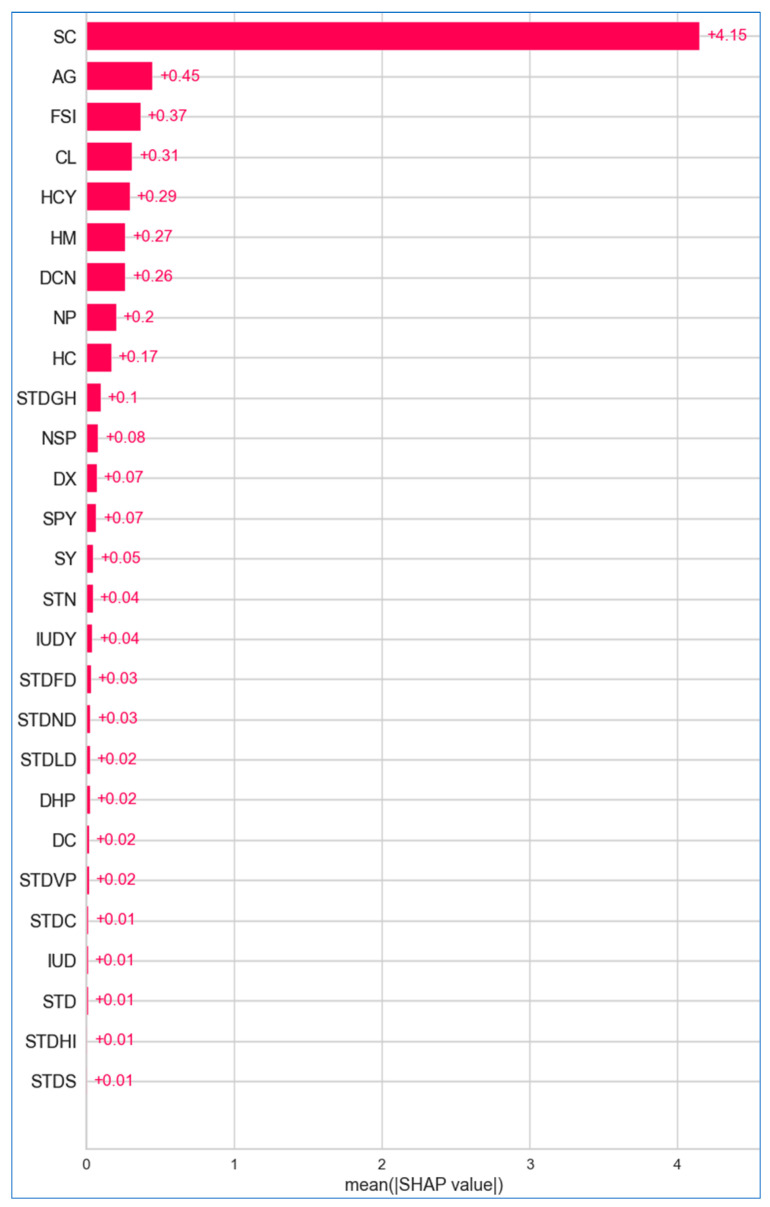
Absolute mean SHAP for cervical cancer.

**Figure 27 bioengineering-12-00472-f027:**
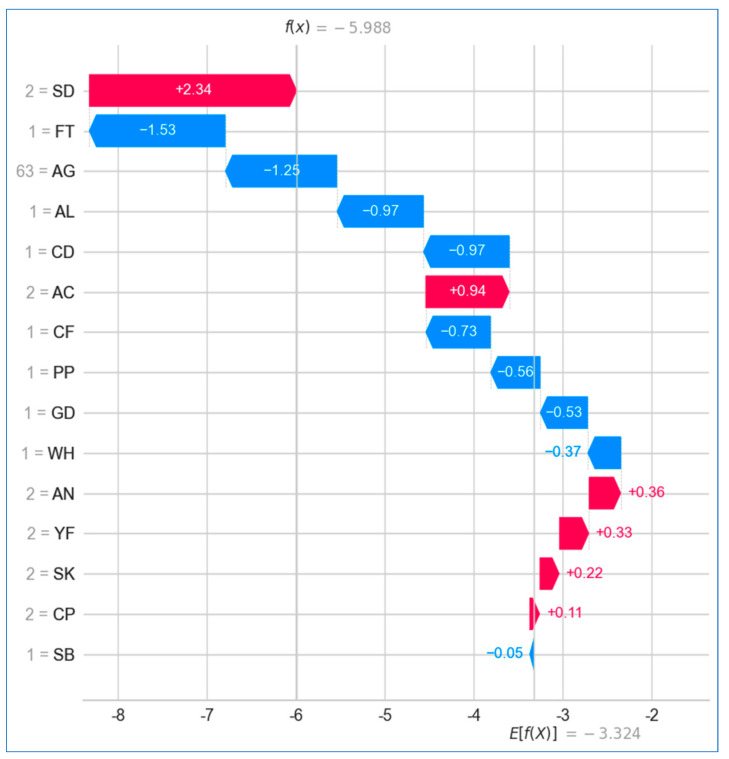
Waterfall plot for lung cancer.

**Figure 28 bioengineering-12-00472-f028:**
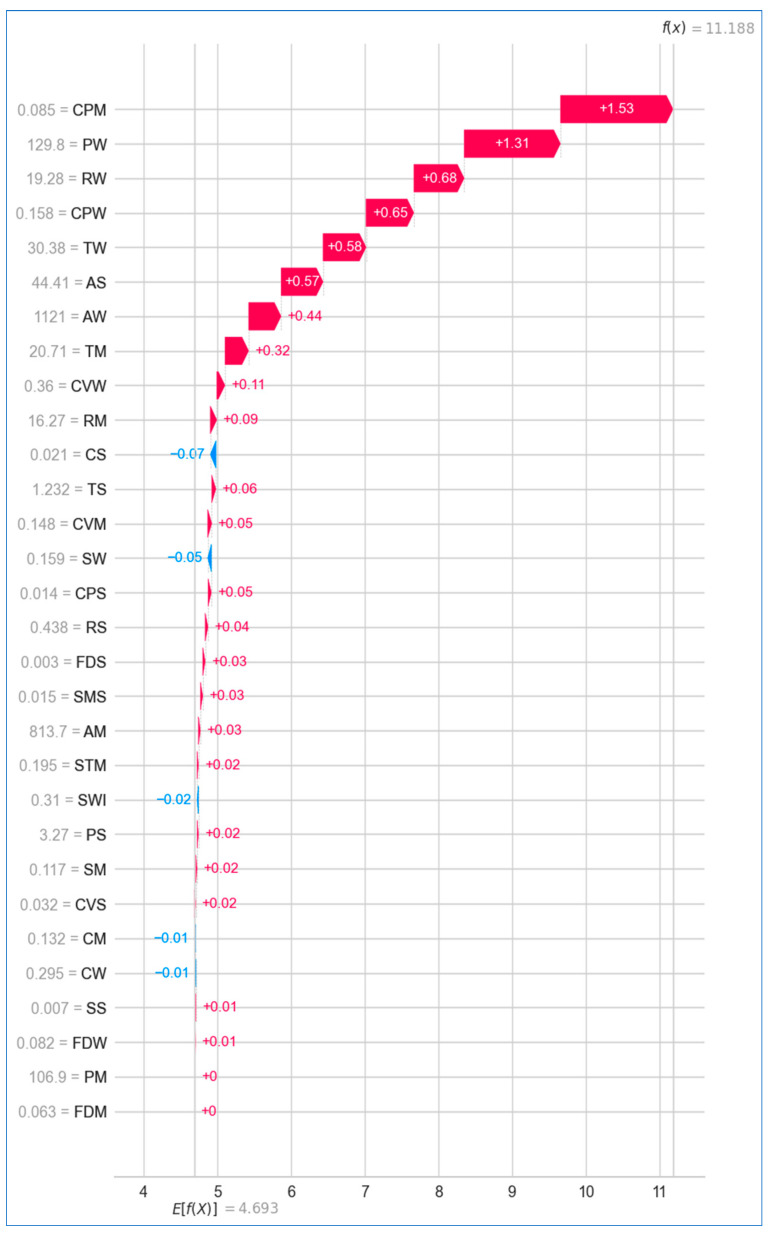
Waterfall plot for breast cancer.

**Figure 29 bioengineering-12-00472-f029:**
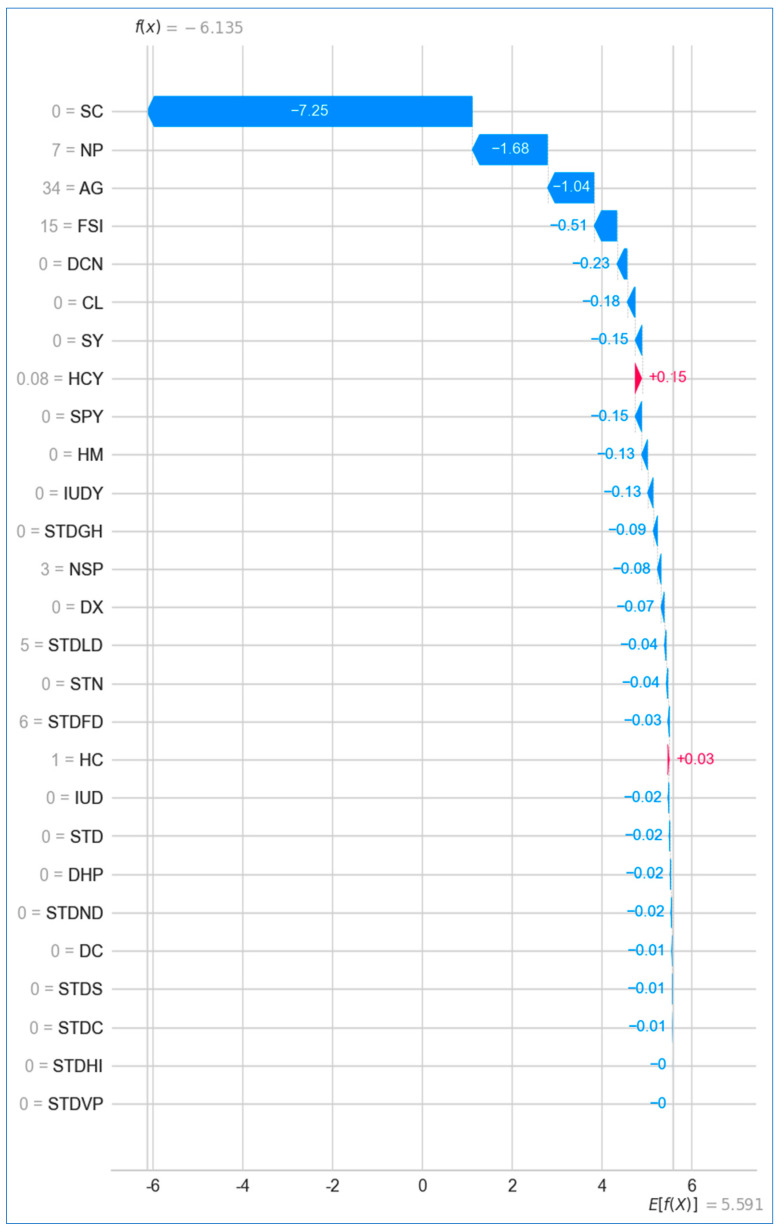
Waterfall plot for cervical cancer.

**Figure 30 bioengineering-12-00472-f030:**
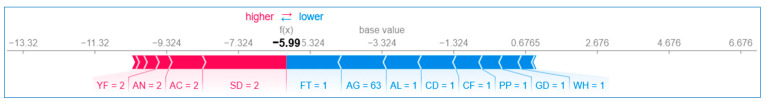
Force plot for lung cancer.

**Figure 31 bioengineering-12-00472-f031:**
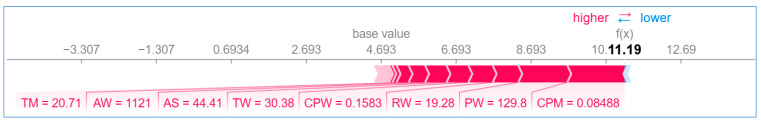
Force plot for breast cancer.

**Figure 32 bioengineering-12-00472-f032:**
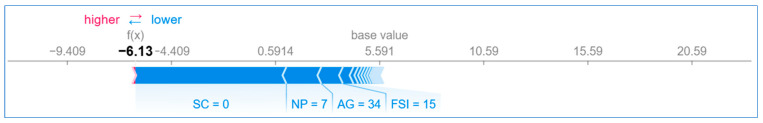
Force plot for cervical cancer.

**Table 1 bioengineering-12-00472-t001:** List of abbreviations.

Abbreviation	Full Name	Abbreviation	Full Name
ADB	AdaBoost	LCPT	Lung cancer prediction tool
AUC	Area under the curve	LDA	Linear discriminant analysis
AUPRC	Area under precision–recall curve	LGBM	Light gradient boosting machine
BCWD	Breast cancer Wisconsin (diagnostic)	LR	Logistic regression
BDT	Boosted decision tree	MB	Multi boosting
BME	Bagging meta-estimator	MCC	Matthews correlation coefficient
BRF	Balanced random forest	MLP	Multilayer perceptron
BSPL	Bootstrap with self-paced learning	NB	Naive Bayes
CB	Cat boosting	NCC	Nearest centroid classifier
CCrf	Cervical cancer (risk factors)	PCA	Principal component analysis
CNN	Convolutional neural network	PDP	Partial dependence plots
CRCB	Clinical research center for breast	RBF	Radial basis function
DF	Decision forest	RF	Random forest
DJ	Decision jungle	RFE	Recursive feature elimination
DNN	Deep neural network	ROC	Receiver operating characteristic
DT	Decision tree	RSF	Random survival forest
ERT	Extremely randomized trees	SD	Standard deviation
ET	Extra trees	SEER	Surveillance, epidemiology, and end results
EXSA	Extended XGB for survival analysis	SHAP	Shapley additive explanations
FN	False negative	SMO	Sequential minimal optimization
FP	False positive	STD	Sexually transmitted diseases
FPR	False positive rate	SVM	Support vector machine
GB	Gradient boosting	TN	True negative
IQR	Interquartile range	TP	True positive
IUD	Intrauterine device	WDBC	Wisconsin diagnostic breast cancer
KNN	K-nearest neighbor	XGB	Extreme gradient boosting

**Table 2 bioengineering-12-00472-t002:** Performance evaluation metrics.

Metric	Significance and Interpretation	Calculation
Accuracy	Accuracy represents the proportion of correct predictions (both true positives and true negatives) out of the total number of predictions. It provides an overall measure of the model’s ability to correctly identify both cancer and non-cancer cases. A high accuracy indicates that the model is performing well, but it may not be the best metric in cases where the data are imbalanced.	$\frac{TP+TN}{TP+TN+FP+FN}$
Precision	Precision measures the proportion of true positive predictions out of all the positive predictions made by the model. It reflects the model’s ability to correctly identify cancer cases without falsely labeling non-cancer cases as cancer. A high precision is crucial in cancer screening to avoid unnecessary further testing or treatment.	$\frac{TP}{TP+FP}$
Recall	Recall, also known as sensitivity, represents the proportion of true positive predictions out of all the actual cancer cases. It reflects the model’s ability to correctly identify cancer cases. A high recall is important in cancer diagnosis to ensure that the model does not miss any cancer cases.	$\frac{TP}{TP+FN}$
F1-score	The F1-score is the harmonic mean of precision and recall, providing a balanced measure of the model’s performance. It is particularly useful when the data are imbalanced, as it considers both the model’s ability to correctly identify cancer cases and its ability to avoid false positives.	$\frac{2\times TP}{2\times TP+FP+FN}$
MCC	MCC is a more comprehensive performance metric that considers all four confusion matrix elements (true positives, true negatives, false positives, and false negatives). It provides a balanced measure of the model’s performance, ranging from −1 to 1, where 1 indicates a perfect prediction, 0 indicates a random prediction, and −1 indicates a completely inverse prediction.	$\frac{TP\times TN-FP\times FN}{\sqrt{\begin{array}{c} \left(TP+FP\right)\times \left(TP+FN\right)\times \\ \left(TN+FP)\times (TN+FN\right) \end{array}}}$
Kappa	Cohen’s Kappa is a statistical measure that evaluates the model’s performance while considering the possibility of chance agreement. It is particularly useful when the data are imbalanced, as it takes into account the expected agreement by chance. Kappa values range from −1 to 1, with 1 indicating perfect agreement, 0 indicating agreement no better than chance, and negative values indicating agreement worse than chance.	$\frac{2\times (TP\times TN-FP\times FN)}{\begin{array}{c} \left(TP+FP\right)\times \left(FP+TN\right)+\\ \left(TP+FN)\times (TN+FN\right) \end{array}}$
ROC curve	The ROC curve is a plot of the true positive rate (sensitivity) against the false positive rate (1—specificity) at various classification thresholds. It provides a visual representation of the trade-off between sensitivity and specificity, allowing you to evaluate the model’s performance across different decision boundaries.	Recall (*y* axis) vs. FPR (*x* axis)
AUC	The AUC of the ROC curve represents the probability that a randomly selected cancer case will be ranked higher than a randomly selected non-cancer case by the model. It ranges from 0 to 1, with 1 indicating a perfect model and 0.5 indicating a random model. A higher AUC value indicates better overall diagnostic performance.	$\int_{0}^{1}TPR({FPR}^{-1}\left(t\right))dt$, t is a threshold

**Table 3 bioengineering-12-00472-t003:** Parameter information of lung cancer dataset.

Parameter	Description	Measurement	Mean	Std	Min	Max
*Demographic and lifestyle parameters*						
Gender (GD)	The gender of the individual	Male/female	0.48	0.50	0	1
Age (AG)	The age of the individual	Numeric	61.96	8.63	21	87
Smoking (SK)	Whether the individual is or has been a smoker.	Yes (2)/no (1)	1.54	0.49	1	2
Alcohol consuming (AC)	Whether the individual consumes alcohol.	Yes (2)/no (1)	1.40	0.49	1	2
Yellow fingers (YF)	Whether the individual has yellowed fingers, often associated with long-term smoking.	Yes (2)/no (1)	1.47	0.49	1	2
*Psychological and social influence factors*						
Anxiety (AN)	Whether the individual suffers from anxiety.	Yes (2)/no (1)	1.42	0.49	1	2
Peer pressure (PP)	Whether the individual feels influenced by peers to engage in risky behaviors like smoking.	Yes (2)/no (1)	1.39	0.48	1	2
*Health and symptom-related parameters*						
Chronic disease (CD)	Whether the individual has any chronic disease.	Yes (2)/no (1)	1.44	0.49	1	2
Fatigue (FT)	Whether the individual experiences fatigue.	Yes (2)/no (1)	1.58	0.49	1	2
Allergy (AL)	Whether the individual has allergies.	Yes (2)/no (1)	1.38	0.48	1	2
Wheezing (WH)	Whether the individual experiences wheezing, which is a high-pitched whistling sound when breathing.	Yes (2)/no (1)	1.40	0.49	1	2
Coughing (CF)	Whether the individual experiences persistent coughing.	Yes (2)/no (1)	1.43	0.49	1	2
Shortness of breath (SB)	Whether the individual experiences shortness of breath.	Yes (2)/no (1)	1.60	0.48	1	2
Swallowing difficulty (SD)	Whether the individual has trouble swallowing.	Yes (2)/no (1)	1.32	0.46	1	2
Chest pain (CP)	Whether the individual experiences chest pain.	Yes (2)/no (1)	1.43	0.49	1	2
*Target variable*						
Lung cancer (LC)	Whether the individual has been diagnosed with lung cancer.	Yes (2)/no (1)	0.50	0.50	0	1

**Table 4 bioengineering-12-00472-t004:** Parameter information of breast cancer dataset.

Parameter	Description	Measurement	Mean	Std	Min	Max
*Mean values (computed on the mean of all cells in the image)*						
Radius_mean (RM)	The mean radius of the cell nuclei, which is the average distance from the center to the boundary of the nucleus.	Decimal	14.12729	3.524049	6.981	28.11
Texture_mean (TM)	The mean standard deviation of the gray-scale values in the image, which measures the smoothness or roughness of the cell nuclei	Decimal	19.28965	4.301036	9.71	39.28
Perimeter_mean (PM)	The mean perimeter of the cell nuclei, representing the boundary length.	Decimal	91.96903	24.29898	43.79	188.5
Area_mean (AM)	The mean area of the cell nuclei, indicating the size of the nuclei.	Decimal	654.8891	351.9141	143.5	2501
Smoothness_mean (SM)	The mean measure of how smooth the edges of the nuclei are, calculated as the variation in radius lengths.	Decimal	0.09636	0.014064	0.05263	0.1634
Compactness_mean (CM)	The mean measure of how compact the nuclei are, computed as PM^2^/AM, indicating the roundness of the nuclei.	Decimal	0.104341	0.052813	0.01938	0.3454
Concavity_mean (CVM)	The mean extent of concave portions of the nucleus boundary.	Decimal	0.088799	0.07972	0	0.4268
Concave points_mean (CPM)	The mean number of concave points on the nucleus boundary.	Decimal	0.048919	0.038803	0	0.2012
Symmetry_mean (SM)	The mean symmetry of the nucleus, measuring how symmetrical the nucleus shape is.	Decimal	0.181162	0.027414	0.106	0.304
Fractal_dimension_mean (FDM)	The mean fractal dimension of the nucleus, which indicates the complexity of the nucleus boundary (calculated as “coastline approximation”).	Decimal	0.062798	0.00706	0.04996	0.09744
*Standard error values (measure of uncertainty/variation in each feature)*						
Radius_se (RS)	The standard error of the radius of the cell nuclei.	Decimal	0.405172	0.277313	0.1115	2.873
Texture_se (TS)	The standard error of the texture, indicating variability in gray-scale intensity.	Decimal	1.216853	0.551648	0.3602	4.885
Perimeter_se (PS)	The standard error of the perimeter of the cell nuclei.	Decimal	2.866059	2.021855	0.757	21.98
Area_se (AS)	The standard error of the area of the cell nuclei.	Decimal	40.33708	45.49101	6.802	542.2
Smoothness_se (SS)	The standard error of the smoothness.	Decimal	0.007041	0.003003	0.001713	0.03113
Compactness_se (CS)	The standard error of the compactness.	Decimal	0.025478	0.017908	0.002252	0.1354
Concavity_se (CVS)	The standard error of the concavity.	Decimal	0.031894	0.030186	0	0.396
Concave points_se (CPS)	The standard error of the concave points.	Decimal	0.011796	0.00617	0	0.05279
Symmetry_se (SMS)	The standard error of the symmetry.	Decimal	0.020542	0.008266	0.007882	0.07895
Fractal_dimension_se (FDS)	The standard error of the fractal dimension.	Decimal	0.003795	0.002646	0.000895	0.02984
*“Worst” values (largest values for each feature)*						
Radius_worst (RW)	The largest radius observed among all cell nuclei in the image.	Decimal	16.26919	4.833242	7.93	36.04
Texture_worst (TW)	The largest texture value observed.	Decimal	25.67722	6.146258	12.02	49.54
Perimeter_worst (PW)	The largest perimeter value observed.	Decimal	107.2612	33.60254	50.41	251.2
Area_worst (AW)	The largest area observed among the nuclei.	Decimal	880.5831	569.357	185.2	4254
Smoothness_worst (SW)	The worst (largest) value of smoothness observed.	Decimal	0.132369	0.022832	0.07117	0.2226
Compactness_worst (CW)	The largest compactness value observed.	Decimal	0.254265	0.157336	0.02729	1.058
Concavity_worst (CVW)	The largest concavity value observed.	Decimal	0.272188	0.208624	0	1.252
Concave points_worst (CPW)	The largest number of concave points observed.	Decimal	0.114606	0.065732	0	0.291
Symmetry_worst (SWI)	The worst (largest) symmetry value.	Decimal	0.290076	0.061867	0.1565	0.6638
Fractal_dimension_worst (FDW)	The largest fractal dimension observed.	Decimal	0.083946	0.018061	0.05504	0.2075
*Target variable*						
Diagnosis (BC)	This is the target variable, indicating whether the tumor is benign or malignant. It represents the classification of the tumor based on the features described above.	1 (yes)/0 (no)	0.372583	0.483918	0	1

**Table 5 bioengineering-12-00472-t005:** Parameter information of cervical cancer dataset.

Parameter	Description	Measurement	Mean	Std	Min	Max
*Demographic and sexual behavior parameters*						
Age (AG)	The age of the patient.	Numeric	27.02395	8.482986	13	84
Number of sexual partners (NSP)	The total number of sexual partners the individual has had, which can influence the risk of contracting STDs.	Numeric	2.535329	1.654044	1	28
First sexual intercourse (FSI)	The age at which the individual had their first sexual intercourse.	Numeric	17.02036	2.805154	10	32
Num of pregnancies (NP)	The total number of pregnancies the individual has had.	Numeric	2.283832	1.408152	0	11
*Smoking-related parameters*						
Smokes (SK)	Whether the individual smokes.	Yes (1)/no (0)	0.147305	0.354623	0	1
Smokes (years) (SY)	The number of years the individual has been smoking.	Numeric	1.249898	4.108449	0	37
Smokes (packs/year) (SPY)	The number of packs of cigarettes the individual smokes per year.	Numeric	0.458571	2.239363	0	37
*Hormonal contraceptive and IUD use*						
Hormonal contraceptives (HC)	Whether the individual uses or has used hormonal contraceptives (e.g., birth control pills).	Yes (1)/no (0)	0.571257	0.495193	0	1
Hormonal contraceptives (years) (HCY)	The number of years the individual has been using hormonal contraceptives.	Numeric	2.26555	3.553566	0	30
IUD (IUD)	Whether the individual uses or has used an intrauterine device (IUD) for birth control.	Yes (1)/no (0)	0.099401	0.299379	0	1
IUD (years) (IUDY)	The number of years the individual has been using an IUD, to quantify long-term use.	Numeric	0.45685	1.83754	0	19
*STD-related parameters*						
STDs (STD)	Whether the individual has had any STDs.	Yes (1)/no (0)	0.094611	0.292852	0	1
STDs (number) (STN)	The total number of STDs the individual has been diagnosed with.	Numeric	0.159281	0.536236	0	4
STDs:condylomatosis (STDC)	Whether the individual has been diagnosed with condylomatosis, a type of genital wart caused by certain strains of HPV.	Yes (1)/no (0)	0.052695	0.223557	0	1
STDs:cervical condylomatosis (STDCC)	A specific condition where condylomatosis affects the cervix.	Yes (1)/no (0)	0	0	0	0
STDs:vaginal condylomatosis (STDVC)	A condition where condylomatosis affects the vaginal area.	Yes (1)/no (0)	0.00479	0.069088	0	1
STDs:vulvo-perineal condylomatosis (STDVP)	Condylomatosis affecting the vulva and perineal areas.	Yes (1)/no (0)	0.051497	0.221142	0	1
STDs:syphilis (STDS)	Whether the individual has had syphilis, another STD that can affect general health and reproductive health.	Yes (1)/no (0)	0.021557	0.145319	0	1
STDs:pelvic inflammatory disease (STDPI)	Whether the individual has had pelvic inflammatory disease, a complication of STDs that affects the female reproductive organs.	Yes (1)/no (0)	0.001198	0.034606	0	1
STDs:genital herpes (STDGH)	Whether the individual has had genital herpes, an STD caused by the herpes simplex virus.	Yes (1)/no (0)	0.001198	0.034606	0	1
STDs:molluscum contagiosum (STDMC)	Whether the individual has been diagnosed with molluscum contagiosum, a viral infection that can be spread sexually.	Yes (1)/no (0)	0.001198	0.034606	0	1
STDs:AIDS (STDAI)	Whether the individual has been diagnosed with AIDS, a condition caused by HIV that weakens the immune system.	Yes (1)/no (0)	0	0	0	0
STDs:HIV (STDHI)	Whether the individual is HIV-positive.	Yes (1)/no (0)	0.021557	0.145319	0	1
STDs:Hepatitis B (STDHB)	Whether the individual has been diagnosed with Hepatitis B, a viral infection that can affect the liver but also has implications for general health.	Yes (1)/no (0)	0.001198	0.034606	0	1
STDs:HPV (STDHP)	Whether the individual has been diagnosed with HPV (human papillomavirus).	Yes (1)/no (0)	0.002395	0.048912	0	1
STDs:Number of diagnosis (STDND)	The number of STD diagnoses the individual has received over time.	Numeric	0.08982	0.306335	0	3
STDs:Time since first diagnosis (STDFD)	The amount of time since the individual’s first STD diagnosis.	Numeric	6.011976	1.70831	1	22
STDs:Time since last diagnosis (STDLD)	The time since the individual’s most recent STD diagnosis.	Numeric	5.069461	1.682884	1	22
*Diagnosis results*						
Dx:CIN (DCN)	Whether the individual has been diagnosed with cervical intraepithelial neoplasia (CIN), a pre-cancerous condition in the cervix.	Yes (1)/no (0)	0.010778	0.10332	0	1
Dx:HPV (DHP)	Whether the individual has been diagnosed with HPV infection.	Yes (1)/no (0)	0.021557	0.145319	0	1
Dx (DX)	A general diagnostic indicator that may represent additional medical conditions diagnosed.	Yes (1)/no (0)	0.028743	0.167182	0	1
*Screening tests*						
Hinselmann (HM)	A result from the Hinselmann test, a colposcopy test used to visually examine the cervix for signs of disease, including cancer.	Yes (1)/no (0)	0.041916	0.200518	0	1
Schiller (SC)	A result from the Schiller test, in which iodine is applied to the cervix to detect abnormalities based on how cells absorb the iodine.	Yes (1)/no (0)	0.087425	0.282626	0	1
Citology (CL)	A cytological examination of cells (e.g., a Pap smear) used to detect abnormal cells that could indicate precancerous or cancerous changes.	Yes (1)/no (0)	0.051497	0.221142	0	1
Biopsy (CC)	A biopsy result, which involves the removal of a small sample of cervical tissue to check for cancerous or pre-cancerous changes.	Yes (1)/no (0)	0.064671	0.246091	0	1
*Target variable*						
Dx:Cancer (DC)	Whether the individual has been diagnosed with cancer.	Yes (1)/no (0)	0.021557	0.145319	0	1

**Table 6 bioengineering-12-00472-t006:** Heat map of the performance evaluation of twelve base models in predicting lung, breast, and cervical cancers.

Metric	Dataset	NB	SVM	QDA	Ridge	KNN	LDA	LR	DT	RF	ET	GB	ADB
Accuracy	Lung	90.28	81.17	87.97	90.28	86.10	93.07	92.60	87.49	89.78	89.76	90.71	89.81
	Breast	94.21	85.65	96.47	95.21	92.20	94.96	94.46	91.93	94.94	96.46	95.69	95.71
	Cervical	88.36	93.33	95.21	95.39	93.50	95.90	94.70	94.87	95.39	94.70	95.56	96.07
Precision	Lung	94.04	89.64	91.21	91.84	88.56	96.07	94.17	91.49	91.73	92.20	92.67	93.26
	Breast	94.08	83.29	95.55	97.90	92.29	98.56	93.79	87.95	93.87	96.60	94.82	94.37
	Cervical	64.69	62.67	68.83	64.00	62.63	65.67	60.00	65.71	69.17	60.17	72.00	81.67
Recall	Lung	95.26	90.00	95.79	97.89	96.84	96.32	97.89	94.74	97.37	96.84	97.37	95.79
	Breast	90.38	84.57	95.14	89.05	86.43	87.71	91.14	91.10	92.43	93.76	93.76	94.43
	Cervical	80.83	78.33	55.83	74.17	71.89	84.17	50.83	58.33	50.83	53.33	64.17	81.50
F1-score	Lung	94.50	86.45	93.28	94.66	92.39	96.08	95.88	92.96	94.37	94.32	94.84	94.31
	Breast	91.95	81.92	95.17	93.03	89.20	92.61	92.40	89.36	93.08	95.02	94.11	94.16
	Cervical	68.01	70.71	79.29	77.25	78.02	73.34	52.81	59.48	55.17	54.94	64.02	82.49
AUC	Lung	92.19	73.21	76.27	61.72	65.06	93.41	92.69	66.54	91.89	87.99	85.99	85.47
	Breast	98.25	89.23	98.58	92.86	95.41	99.09	99.00	91.75	98.41	98.97	98.59	98.28
	Cervical	92.86	97.24	97.36	93.21	85.34	96.60	93.41	77.89	97.55	95.73	97.34	98.23
Kappa	Lung	50.39	4.68	30.18	38.75	8.85	64.76	59.03	35.25	38.97	40.35	45.65	41.40
	Breast	87.43	70.40	92.39	89.41	83.10	88.82	88.04	82.87	89.10	92.27	90.72	90.77
	Cervical	42.78	18.91	56.86	64.86	53.81	71.19	50.28	56.84	53.28	52.36	61.79	63.52
MCC	Lung	51.92	5.71	31.51	42.34	9.81	67.10	63.04	37.50	42.06	43.35	49.55	43.43
	Breast	87.73	72.54	92.60	89.86	83.28	89.37	88.11	83.06	89.17	92.46	90.92	91.05
	Cervical	47.95	20.90	58.46	65.86	51.87	72.07	51.43	58.20	55.53	53.23	63.82	65.67

**Table 7 bioengineering-12-00472-t007:** Ranking of the base learners based on their accuracy in predicting the three cancer diseases.

Cancer Type	1st	2nd	3rd	4th	5th	6th
Lung	NB	Ridge	LDA	LR	GB	ADB
Breast	QDA	Ridge	LDA	ET	GB	ADB
Cervical	QDA	Ridge	LDA	RF	GB	ADB

**Table 8 bioengineering-12-00472-t008:** Hyperparameter values for the constituent top six models (for each dataset) of the stacking model.

Algorithm	Hyperparameters
NB	random_state = None, c = 100, gamma = 0.001, kernel = rbf, probability = False, verbose = False, refit = True, verbose = 3
QDA	reg_param: 0.2, estimator = qda, param_grid = param_grid, cv = 5, scoring = ‘accuracy’
Ridge	alpha = 1.0, fit_intercept = True, copy_X = True, max_iter = None, tol = 0.0001, solver = ‘auto’, positive = False, random_state = None
LDA	estimator = lda, param_grid = param_grid, cv = 10, scoring = accuracy, solver = lsqr, shrinkage = ‘auto’
LR	param_grid = param_grid, C = 0.01, penalty: elasticnet, solver = linear, max_iter = 200, l1_ratio = 0.5, verbose = True, n_jobs = −1
RF	n_estimators = 1000, criterion = ‘gini’, max_depth = None, min_samples_split = 2, min_samples_leaf = 1, max_features = 16, bootstrap = True, random_state = 42
ET	n_estimators = 1000, criterion = ‘gini’, max_depth = 1000, min_samples_split = 10, min_samples_leaf = 2, max_features = 10, bootstrap = 2, random_state = 42
GB	random_state = 45, learning_rate = [0.1, 2, 5], n_estimators = 5000, max_depth = 4, weight = 6, verbose = 1
ADB	random_state = 45, learning_rate = [0.01, 0.05], n_estimators = 200, algorithm = ‘SAMME.R’, n_jobs = n_jobs

**Table 9 bioengineering-12-00472-t009:** Confusion matrices of the proposed stacking models.

Dataset	Model	TP	FP	FN	TN
Lung cancer	Random	86	0	2	74
	Top 6	87	1	0	74
Breast cancer	Random	96	2	2	111
	Top 6	94	0	1	116
Cervical cancer	Random	210	0	2	236
	Top 6	214	1	1	232

**Table 10 bioengineering-12-00472-t010:** Comparative analysis with similar research works.

Reference	Considered Models	Dataset Used	Sample Size	No. of Features	Splitting Ratio (Train:Test)	Highest Accuracy (%)	Recall (%)	Precision (%)	F1-Score (%)	MCC (%)	Kappa	AUC (%)	XAI
Abdullah et al. [13]	SVM, KNN, and CNN	Lung cancer dataset UCI machine learning repository	32	56	-	95.56 (SVM)	95.60	95.60	95.40	-	-	95.50	×
Safiyari and Javidan [26]	Bagging, dagging, ADB, MB and random subspace with a combination of several base classifiers	SEER data (released in April 2016)	643,924	149	-	88.98 (ADB)	-	-	-	-	-	94.90	×
Wang et al. [27]	Bagging, RF with BSPL	US National Library of Medicine National Institutes of Health	720	3	70:30	97.96 (RFSPL)	-	-	98.11	-	-	98.21	×
Siddhartha et al. [28]	Bagging with RF	Wroclaw Thoracic Surgery Centre for Patients (2007 to 2011)	470	17	90:10	84.04	94.28	85.71	89.79	-	-	83	×
Ahmad and Mayya [25]	RF for LCPT	International cancer database and local medical questionnaire	1000	23	-	93.33 (RF)	100	81.82	90.09	93.33	84.28	100	×
Mamun et al. [29]	XGB, LGBM, bagging, and ADB	Lung cancer dataset by staceyinrobert, Data World	309	16	70:30	94.42 (XGB)	94.46	95.66	-	-	-	98.14	×
Patra [14]	NB, KNN, and RBF	Lung cancer UCI machine learning repository	32	56	80:20	81.25 (RBF)	81.30	81.30	81.30	-	-	74.90	×
Zhang et al. [38]	LR, NB, RF, and XGB	UK Biobank	467,888	49	70:30	98.93 (XGB)	98.13	99.72	98.92	97.90	97.90	99.80	×
Radhika et al. [56]	DT, SVM, NB, and LR	Lung cancer UCI machine learning repository (D1) and Data World (D2)	32 (D1), 1000 (D2)	57 (D1), 25 (D2)	-	99.20 (SVM)	99.01	89.99	99	89.36	-	99	×
Jaiswal et al. [30]	SVM, LR, KNN, NB, DT, RF, ADB, XGB, and I-XGB	BCWD	569	30	80:20	99.84 (I-XGB)	100	99.50	98.50	-	-	-	×
Rabiei et al. [31]	RF, GBT, MLP, and GA	Motamed Cancer Institute (ACECR) collected from Tehran, Iran	5178	24	75:25	80 (RF)	95	84.82	89.62	78.57	78	59	×
Nemade and Fegade [20]	NB, LR, SVM, KNN, DT, RF, ADB, and XGB	BCWD	569	30	80:20	97 (XGB)	96	98	97	-	-	99	×
Naji and Filali [32]	SVM, RF, LR, DT, and KNN	BCWD	569	30	75:25	97.20 (SVM)	94	98	97.93	-	-	96.60	×
Fu et al. [33]	EXSA, Cox, RSF, and GBM	Clinical Research Center for Breast (CRCB) collected from China hospital	12,119	20	70:30	83.45 (Cox)	-	-	-	-	-	83.85	×
Uddin et al. [23]	SVM, RF, KNN, DT, NB, LR, ADB, GB, MLP, NCC, and voting	CCrf, UCI machine learning repository	858	36	80:20	99.16 (voting)	100	100	100	-	-	-	×
Alam et al. [34]	BDT, DF, and DJ	CCrf, UCI machine learning repository	858	36	-	94.10 (BDT)	89.60	89.60	89.60	-	86.90	97.80	×
Song et al. [35]	LR, DT, SVM, MLP, KNN, and voting	Collected dataset (D1), and CCrf, UCI machine learning repository (D2)	472 (D1), 858 (D2)	50 (D1), 36 (D2)	-	83.16 (voting)	28.35	51.73	32.80	34.92	66.82	-	×
Jahan et al. [36]	MLP, RF, KNN, DT, LR, SVC, GBC and ADB	CCrf, UCI machine learning repository	858	36	70:30	98.10 (MLP)	98	98	98	95.22	95.22	97.61	×
Alsmariy et al. [37]	LR, DT, RF, and voting	CCrf, UCI machine learning repository	858	36	-	98.49 (voting)	98.60	95.16	98.37	58.90	72.50	99.80	×
Ali et al. [24]	RF, SVM, GNB, DT, and ensemble	Sobar-72 (D1), and CCrf, UCI machine learning repository (D2)	72 (D1), 858 (D2)	19 (D1), 36 (D2)	70:30	98.06 (ensemble)	100	98.77	98.97	98	98	97	√
Aravena et al. [39]	XGB, LR, RF, and SVM	Breast cancer patients from Indonesia	400	7	75:25	85 (XGB)	85.40	79.50	-	-	-	-	√
Makubhai et al. [40]	DT, LR, RF, and XGB	Five datasets (D1, D2, D3, D4, D5)	500 (D1), 1000 (D2), 750 (D3), 2000 (D4), 1500 (D5)	20 (D1), 15 (D2), 25 (D3), 30 (D4), 18 (D5)	70:30	95.5 (XGB)	-	-	-	91	91	-	√
This paper	LR, NB, KNN, SVM, LDA, CB, GB, XGB, ADB, LGBM, DT, RF, ET, BDT, BME	Lung cancer dataset by staceyinrobert, Data World	309	16	70:30	99.22 (stacking)	93.16	100	96.40	96.07	95.96	98.23	√
		BCWD	569	31		98.85 (stacking)	99.51	98.88	99.19	97.20	97.19	99.93	
		Cervical cancer	835	36		99.78 (stacking)	100	99.76	99.88	98.31	98.29	99.39	

## Data Availability

The datasets used in the experiment are publicly available at the following addresses: https://www.kaggle.com/datasets/mysarahmadbhat/lung-cancer (accessed on 12 February 2025), https://www.kaggle.com/datasets/uciml/breast-cancer-wisconsin-data (accessed on 12 February 2025), and https://www.kaggle.com/datasets/ranzeet013/cervical-cancer-dataset (accessed on 12 February 2025). The code used in this study is openly available on GitHub at the following repository: https://github.com/Shahid92-Phd/Multi-Cancer (accessed on 24 April 2025).

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
