# Peer review of "Enhanced and Interpretable Prediction of Multiple Cancer Types Using a Stacking Ensemble Approach with SHAP Analysis"

_bioengineering, 2025, doi:10.3390/bioengineering12050472_

Round 1
Reviewer 1 Report
Comments and Suggestions for Authors
The article touches upon an important and relevant topic of cancer diagnostics.
However, although the article is large, there are a number of significant comments.
- The title of the article seems not entirely accurate. The article talks about the applicability of models to predicting different types of cancer, since training is carried out on 3 different data sets. Most likely, this is not Multi-Cancer Prediction.
- The entire article does not contain references to sources from the list of references. There are none in Table 6, which provides a comparative analysis of the proposed approach and already known ones.
- The article, despite its large volume, has a descriptive superficial nature. Its results are problematic to reproduce. Formulas are given only for quality metrics in Table 1.
- The theoretical part is missing in the article. Right after the section "2. Related Work " there is the section "3. Results", which has some theoretical considerations built in.
- Although Figure 1 is not a diagram of the algorithm, the exit from the "diamond" block looks very strange. The same question applies to the "diamond" blocks in Figure 5. Usually the entrance to the "diamond" is performed from the top, and the exits, for example, are to the right and down.
- Although the decoding of the names of classifiers, metrics, and features is given at the end of the article in the appendices, I would like to see it at the first mention. The text should also say where to look for the names of features when we see their abbreviated names on the graphs. It seems that some full names of features do not match the first letters in the abbreviations.
- On page 9, line 347, the figure number is not indicated.
- Why does section 5.2 talk about section 5.3.1 (line 359).
- The authors talk about removing insignificant features in 2 data sets. And what do the authors do with these features for the breast cancer set? This is page 11.
- The title of table 5 talks about 6 models, although the table itself has 9 rows (9 models).
- The authors say that table 5 is an error matrix. But this is not true. There is no error matrix.
- The colors in figures 10-12 are almost indistinguishable, they need to be changed. In figure 11, all the colors have merged into one.
- It is unclear how the selection of 6 random classifiers was carried out. What if they accidentally coincide with the best ones?
- It is strange that grid search gave better results. What step for each parameter did the authors use? How many iterations were there in random search? How much time was spent on searching in each of the methods for selecting optimal hyperparameter values? It would make sense to use a population optimization algorithm for searching.
- The description of SHAP Analysis is not complete, although the authors provide many graphs, but use SHAP Analysis without explicitly indicating the formulas.
- The stacking process is not described fully enough.
The English could be improved to more clearly express the research..
Author Response
The article touches upon an important and relevant topic of cancer diagnostics. However, although the article is large, there are a number of significant comments.
- The title of the article seems not entirely accurate. The article talks about the applicability of models to predicting different types of cancer, since training is carried out on 3 different data sets. Most likely, this is not Multi-Cancer Prediction.
Response: We sincerely appreciate the reviewer's concern. The term "multi-cancer prediction" was intended to convey the model’s versatility across distinct cancer types (lung, breast, cervical) rather than simultaneous prediction. To enhance clarity, we have revised the title to: “Enhanced and Interpretable Prediction of Multiple Cancer Types Using a Stacking Ensemble Approach with SHAP Analysis.” By considering three cancer types, our aim was to generalize our designed model for different cancers. However, we acknowledge that other cancer types could also be considered. Given that our model demonstrated satisfactory performance across these three cancer types, we are hopeful it will also be effective for other cancers.
- The entire article does not contain references to sources from the list of references. There are none in Table 6, which provides a comparative analysis of the proposed approach and already known ones.
Response: We sincerely apologize for any confusion that may have arisen. We would like to confirm that all references, including those in Table 6 (comparative analysis), are indeed cited in the text. It is possible that the issue may have occurred due to formatting during the submission process. We will make sure to carefully review the final version of the manuscript to ensure that all references are clearly visible. Additionally, we kindly suggest that the journal might consider verifying the output from the submission system. Thank you for your understanding and cooperation.
- The article, despite its large volume, has a descriptive superficial nature. Its results are problematic to reproduce. Formulas are given only for quality metrics in Table 1.
Response: We appreciate the reviewer's observation regarding the length of the paper. We intentionally included detailed sections to ensure clarity for readers from diverse backgrounds. For example, researchers in medical fields may not be familiar with ensemble learning methodologies or XAI, so a brief foundational discussion on these topics could enhance the article's accessibility and understanding for a broader audience. Additionally, we conducted an extensive analysis of the results to ensure that the functioning and effectiveness of the designed model are clearly conveyed. The complete code for this study is available on GitHub (https://github.com/Shahid92-Phd/Multi-Cancer), which we hope will assist interested readers in reproducing the experiment. Formulas are provided where necessary, such as for the scaling method (Eq. 1) and SHAP (Eq. 2).
- The theoretical part is missing in the article. Right after the section "2. Related Work " there is the section "3. Results", which has some theoretical considerations built in.
Response: We sincerely appreciate your attention to the oversight regarding the section title. We have now updated the title to "Research Methodology," which encompasses the theoretical considerations crucial for understanding the foundation of our work. This section now thoughtfully outlines the theoretical background of the methodologies employed, including the principles of stacking ensemble models.
- Although Figure 1 is not a diagram of the algorithm, the exit from the "diamond" block looks very strange. The same question applies to the "diamond" blocks in Figure 5. Usually the entrance to the "diamond" is performed from the top, and the exits, for example, are to the right and down.
Response: We sincerely appreciate the feedback regarding the figure formatting. In response to the suggestion, we have made revisions to Figure 1 to enhance its clarity. Specifically, we have adjusted the flow of the diagram, ensuring that the entrance to decision blocks (diamonds) is from the top, with exits clearly marked to the right or downward. However, we have opted to keep Figure 5 unchanged, as reorienting the diagram would result in increased length. While we acknowledge that it is generally preferred for the incoming arrow in a diamond box (decision point) to be placed at the top of the symbol, we believe it is not strictly necessary. The diamond symbol can be oriented with the arrow entering from any side, provided the direction of flow is clearly indicated. Our primary aim is to ensure the diagram is clear and easy to follow, rather than adhering rigidly to a specific orientation.
- Although the decoding of the names of classifiers, metrics, and features is given at the end of the article in the appendices, I would like to see it at the first mention. The text should also say where to look for the names of features when we see their abbreviated names on the graphs. It seems that some full names of features do not match the first letters in the abbreviations.
Response: We sincerely appreciate the reviewer's observation regarding the typical placement of the abbreviation table at the beginning of the paper, a convention we generally follow. However, in the Bioengineering journal, the standard practice is to include the abbreviation table in the appendix, as per the author guidelines. Considering your suggestion, we have placed the abbreviation table (Table 1) in the beginning of the manuscript.
- On page 9, line 347, the figure number is not indicated.
Response: We sincerely apologize for this oversight. We have ensured that the figure number is now correctly indicated as Fig. 4 in the revised manuscript: “When considering the average accuracy across all three datasets, as illustrated in Fig. 4, LDA was the top performer with an average accuracy of 94.64%, followed by GB (93.99%) and LR (93.92%). The performance of the other models, based on different evaluation metrics, can be interpreted from the figure.” We have meticulously double-checked all figure references to ensure consistency and accuracy throughout the document, aiming to prevent any future discrepancies.
- Why does section 5.2 talk about section 5.3.1 (line 359).
Response: We are grateful to the reviewer for highlighting this oversight. It should indeed be 5.1, and we have now made the necessary correction.
- The authors talk about removing insignificant features in 2 data sets. And what do the authors do with these features for the breast cancer set? This is page 11.
Response: We sincerely appreciate your thoughtful question regarding feature elimination. We decided to remove certain features from the cervical dataset because we observed that seven out of the 33 features did not contribute to the prediction. In the case of the breast cancer dataset, after conducting Recursive Feature Elimination (RFE), we found that while some features had a lesser impact on the model’s overall performance, none had a zero or negligible contribution, unlike in the cervical cancer dataset. Therefore, we chose to retain all features for the breast cancer dataset. Regarding lung cancer, we found that all features were significant.
- The title of table 5 talks about 6 models, although the table itself has 9 rows (9 models).
Response: We sincerely apologize for any lack of clarity on our part. The reviewer has rightly pointed out the mismatch. We would like to kindly clarify that we considered the top six models for each dataset. While most of the models are common across the three datasets, there are also some unique models that appear in the top list of certain datasets. Consequently, there is a total of nine models.
- The authors say that table 5 is an error matrix. But this is not true. There is no error matrix.
Response: We sincerely appreciate the reviewer's keen observation in identifying the mistake. It appears that the hyperparameter table was mistakenly repeated in Table 5 instead of the intended confusion matrix. We have now addressed this issue, and the confusion matrix has been appropriately included. Thank you for your understanding and support.
- The colors in figures 10-12 are almost indistinguishable, they need to be changed. In figure 11, all the colors have merged into one.
Response: We sincerely appreciate the reviewer's observation regarding the readability of the figures in normal view. Due to the program-generated nature of these figures, we face some limitations in modifying them. However, we kindly suggest that zooming in might enhance their visibility. Thank you for your understanding.
- It is unclear how the selection of 6 random classifiers was carried out. What if they accidentally coincide with the best ones?
Response: We sincerely appreciate the reviewer's concern regarding the selection of six random classifiers. To ensure a comprehensive and robust analysis, we employed three categories of models: base models, Bagging, and Boosting. We selected two models from each category using the `random()` function. Through careful experimentation with various permutations and combinations by executing the `random()` function repetitively, we identified the optimal combination: SVM, KNN, DT, ET, GB, and ADB. This selection is not arbitrary; it is the result of a deliberate and calculated process designed to maximize performance and accuracy. This information is now included in Section 5.2.
- It is strange that grid search gave better results. What step for each parameter did the authors use? How many iterations were there in random search? How much time was spent on searching in each of the methods for selecting optimal hyperparameter values? It would make sense to use a population optimization algorithm for searching.
Response: We sincerely appreciate the reviewer's insightful observation. While it is anticipated that random search may surpass grid search in certain applications, in our specific case, grid search demonstrated superior results concerning performance evaluation metrics for the models considered in multi-cancer diseases. We diligently explored various permutations and combinations to determine the optimal hyperparameter values for grid search, taking into account our computational resources. For random search, we conducted 50 iterations per model, utilizing randomized combinations drawn from broader distributions of the same hyperparameters. We are grateful for the reviewer’s suggestion. Indeed, optimization techniques such as Genetic Algorithms (GA) or Particle Swarm Optimization (PSO) may offer better exploration of the hyperparameter space. We plan to explore these approaches in future work to evaluate their effectiveness compared to traditional search methods.
- The description of SHAP Analysis is not complete, although the authors provide many graphs, but use SHAP Analysis without explicitly indicating the formulas.
Response: We greatly appreciate the reviewer's insightful suggestion. Accordingly, we have incorporated the equation for calculating SHAP values in Section 6.2.1.
- The stacking process is not described fully enough.
Response: We sincerely appreciate your valuable observation. The theoretical framework of stacking is detailed in Section 3.2, while its operational principle is depicted in Figure 2. Furthermore, Section 5.2 offers a comprehensive description of the designed stacking models. We have also included the meta-learner utilized in the stacking process (Section 5.2), which was previously omitted.
- The English could be improved to more clearly express the research.
Response: We thank the reviewer 1 for this valuable suggestion. During revision, we have carefully reviewed the manuscript to address any grammatical errors. Additionally, we have utilized editing tools such as Grammarly to ensure the sentence structure is correct and to enhance the overall quality of the writing. We will further enhance it during proofing stage.
Reviewer 2 Report
Comments and Suggestions for Authors
The authors evaluated a method for combining machine learning models to classify individuals with cancer. They also applied a well-known explainability library to support the interpretation of the outcomes. The results are interesting; however, several aspects of the presentation require revision.
- Acronyms should be presented in full before being used.
- Many parts of the introduction are classic machine learning project challenges (eg lines 46 to 56 and others). Perhaps some space could be devoted to motivating the use of such tools in the field of health, such as:
- https://doi.org/10.1109/JBHI.2021.3072628
- https://doi.org/10.1145/3469089
- https://doi.org/10.1145/3421937.3421970
- https://doi.org/10.3390/healthcare10030422
- https://doi.org/10.1109/JPROC.2018.2791463
- https://doi.org/10.1016/j.media.2021.102125
- Line 72 “black-box”. This appellation is usually used for neural networks, some of the basic (level-0) models used by the authors are inherently explainable.
- “Rigorous evaluation of the models using various performance metrics, including accuracy, recall, precision, F1-score, AUC, Kappa, and MCC statistics.” This is not a contribution, but a minimum requirement.
- Section 2. All bibliographical references were skipped.
- A space should not be placed before commas or full stops.
- Section 3. Check the title.
- Line 240 “We initially implemented twelve prediction models”. Implemented from scratch (why?) or were libraries used?
- The presentation section of the datasets should be brought forward, moreover, the balancing activity seems to play a non-negligible role, how did it impact the results? Going, in the case of the cervical dataset from 54 to 710 is very impactful.
- Line 246 “six randomly picked learners” Why random?
- The stacking approach is not proposed by the authors; it should be clarified.
- Lines 273-279. Which model was used for level-1?
- Figure 11 and similar are of little use, the significant portion of the chart without a zoom is unreadable.
Author Response
The authors evaluated a method for combining machine learning models to classify individuals with cancer. They also applied a well-known explainability library to support the interpretation of the outcomes. The results are interesting; however, several aspects of the presentation require revision.
- Acronyms should be presented in full before being used.
Response: We sincerely appreciate the reviewer's observation regarding the typical placement of the abbreviation table at the beginning of the paper, a convention we generally follow. However, in alignment with the standard practice of the Bioengineering journal, the abbreviation table is placed at the end of the manuscript, and we adhered to the author guidelines accordingly. Based on your valuable suggestion, we have now positioned the table (Table 1) in the beginning of the manuscript.
- Many parts of the introduction are classic machine learning project challenges (eg lines 46 to 56 and others). Perhaps some space could be devoted to motivating the use of such tools in the field of health, such as:
- https://doi.org/10.1109/JBHI.2021.3072628
- https://doi.org/10.1145/3469089
- https://doi.org/10.1145/3421937.3421970
- https://doi.org/10.3390/healthcare10030422
- https://doi.org/10.1109/JPROC.2018.2791463
- https://doi.org/10.1016/j.media.2021.102125
Response: We sincerely appreciate your insightful comments and suggestions regarding the motivations of our work. In the second paragraph of the Introduction section, we discuss the challenges associated with traditional machine learning algorithms in healthcare and clinical applications, particularly in relation to cancer predictions. The third paragraph highlights the importance of the interpretability of prediction models within the context of disease prediction, specifically cancer.
We are grateful to for recommending relevant literature. We have carefully reviewed all the suggested papers; however, we found that none are directly pertinent to our study. Most of the papers applied neural networks to image or signal data, and none reported on ensemble learning. If this reviewer still prefers us to cite some, or other more related papers, we will evaluate and do it in the next version.
- Line 72 “black-box”. This appellation is usually used for neural networks, some of the basic (level-0) models used by the authors are inherently explainable.
Response: We sincerely appreciate the reviewer's valuable observation. We acknowledge that the term “black-box” is traditionally linked with complex models such as deep neural networks. While some level-0 models in our study—such as LR, NB, and SVM—are inherently interpretable, our stacking ensemble combines these models in a manner that enhances overall complexity and diminishes direct interpretability. Consequently, we employed the term "black-box" to refer to the overall ensemble model rather than the individual base learners.
- “Rigorous evaluation of the models using various performance metrics, including accuracy, recall, precision, F1-score, AUC, Kappa, and MCC statistics.” This is not a contribution, but a minimum requirement.
Response: We sincerely appreciate your feedback. We concur that conducting a thorough evaluation using a variety of performance metrics is essential for model assessment. However, we would like to gently point out that many studies often report only a limited selection of metrics. In our work, we have endeavored to include a comprehensive set of metrics—accuracy, recall, precision, F1-score, AUC, Kappa, and MCC statistics—to provide a more nuanced understanding of model performance. This is why we have included it in our list of contributions. We are grateful for your insights and hope this explanation clarifies our rationale.
- Section 2. All bibliographical references were skipped.
Response: We regret to inform you that we were unable to identify the issue. We conducted a thorough review of the references and confirmed that all the papers listed in the reference section are cited within the text. It is possible that the version of the manuscript received by the reviewer from the journal contained discrepancies in the references. We appreciate your understanding and are here to assist further if needed.
- A space should not be placed before commas or full stops.
Response: We sincerely appreciate your observation. After conducting a thorough review of the manuscript, we did not identify any such errors. It is possible that these issues may have arisen during the generation of the review-only format of the manuscript. However, please rest assured that such errors are typically addressed during the proofing stage.
- Section 3. Check the title.
Response: We sincerely acknowledge a minor oversight in our document. It appears that Section 3 was mistakenly titled "Results" instead of "Research Methodology." We have duly addressed this issue, and we are grateful for the reviewer's keen attention to this detail.
- Line 240 “We initially implemented twelve prediction models”. Implemented from scratch (why?) or were libraries used?
Response: We express our sincere gratitude to the reviewer for their thorough inquiries. The twelve prediction models were not developed entirely from the ground up. Instead, we employed established machine learning libraries, specifically Scikit-learn (sklearn), for model implementation. Additionally, we utilized PyCaret, an open-source, low-code machine learning library, to automate the workflows of the machine learning models under consideration. These libraries provide optimized, well-tested versions of algorithms, allowing us to focus on model tuning, evaluation, and integration within the stacking ensemble, rather than on low-level algorithmic development.
- The presentation section of the datasets should be brought forward, moreover, the balancing activity seems to play a non-negligible role, how did it impact the results? Going, in the case of the cervical dataset from 54 to 710 is very impactful.
Response: We sincerely appreciate your suggestion. We have now relocated the dataset tables to the relevant section (Section 4) from the Appendix.
The application of SMOTE-based oversampling played a crucial role in addressing class imbalance, particularly within the cervical cancer dataset, where the minority class was notably underrepresented. By increasing the positive class instances from 54 to 710, we ensured that the learning algorithm had sufficient examples to identify relevant patterns, thereby mitigating bias towards the majority class. We observed a significant enhancement in the evaluated metrics, particularly recall and F1-score, in the cervical cancer predictions following the balancing process. Prior to balancing, our models demonstrated satisfactory accuracy and precision but exhibited poor recall, indicative of numerous false negatives. Following the application of SMOTE, recall improved substantially (e.g., up to 99.78% in the stacking model), which is crucial in a medical context where false negatives can have critical implications.
- Line 246 “six randomly picked learners” Why random?
Response: We would like to express our heartfelt gratitude to the reviewer for their thorough and insightful observations. In our study, we conducted multiple executions of the random() function, utilizing a total of six models. Our objective was to assess the performance of a stacking model composed of a diverse set of base learners, which were not selected based on prior performance, to establish a baseline for comparison against a performance-optimized stacking model. This approach underscores the importance of selecting base learners based on empirical performance, as demonstrated by the second stacking model constructed with the top six learners.
- The stacking approach is not proposed by the authors; it should be clarified.
Response: We appreciate the reviewer's observation regarding the stacking approach, acknowledging that it is indeed a well-established method. However, we would like to highlight that we have developed a unique stacking model, which has shown superior performance in predicting three types of cancer.
- Lines 273-279. Which model was used for level-1?
Response: We express our sincere gratitude for your insightful question. In our stacking framework, for the level-1 (meta) model, we chose to employ the SVM model for all diseases using the first approach (random base models). In the second approach (best base models), we selected the LR model for lung cancer and Ridge for breast and cervical cancer, respectively. This decision was guided by the simplicity, interpretability, and robust generalization performance of these models, especially when integrating the outputs of diverse base learners. These meta-learners also tend to perform admirably with the meta-features derived from the probabilistic outputs of the base models. We have now included this information in Section 5.2.
- Figure 11 and similar are of little use, the significant portion of the chart without a zoom is unreadable.
Response: We appreciate the reviewer's insight regarding the readability of Figure 11 in its standard view. The figures are generated by the program, which somewhat limits our ability to modify them extensively. However, we kindly suggest that enhanced visualization can be achieved by zooming in.
Reviewer 3 Report
Comments and Suggestions for Authors In this study, the authors develop a “stacking ensemble” approach for predicting lung, breast, and cervical cancers with lifestyle and clinical data, focusing on explainability with SHAP analysis. A total of 12 base learners were used, and the “stacking” model, which is a combination of the 6 best models, was tested on three different datasets. Both the accuracy and explainability of the model are highlighted. The authors achieved high metrics such as 99.28% accuracy, 99.55% precision, 97.56% recall, and 98.49% F1-score for three cancer types. While most studies focus on only one type of cancer, in this study, three different types were considered, which can be considered as a significant contribution. With SHAP analysis, the decision mechanisms of the model are made comprehensible, and two different stacking approaches are tested and compared with random and performance-selected models. The study achieved higher accuracy and AUC than other methods in the literature. However, I would kindly suggest the following suggestions to the authors.1- The lung cancer data contains only 309 samples, which is a problem in terms of generalizability, and I suggest that it be expanded.
2- The data are taken from open sources, but geographical and demographic diversity is limited; this should be explained in detail, and this situation should be resolved.
3- Only SMOTE is used, there is no comparison with different balancing techniques (ADASYN, Tomek Links), other comparisons should be included.
4- SHAP images are not detailed enough; also, clinical interpretation is missing.
5- It was not tested with real-world data, and no evaluation of clinical integration was presented.
6- Table 1 Evaluation metrics are well explained, Table 2 has a detailed comparison of 12 basic models; however, it is too dense and could have been better grouped.
7- Confusion matrices are presented in Table 5, but they should be supported with clear visualizations.
8- Figure 4-9: Performance comparisons are detailed, but axis names should be written more clearly in some graphs. Figure 10-12: AUC-ROC curves are useful, but visual resolution could be improved.
9- Figure 13-20: Comparisons for each metric are detailed. However, some figures repeat each other and should be simplified.
10- The transferability of the model is unclear, and the applicability to different datasets or populations has not been tested.
11- In particular, data preprocessing, missing data management, and feature selection should be presented in more detail.
12- Code sharing is important for the reproducibility of the work. Also, optimization was done with high precision, but the recall was low. It may miss important cases in the real world.
Author Response
In this study, the authors develop a “stacking ensemble” approach for predicting lung, breast, and cervical cancers with lifestyle and clinical data, focusing on explainability with SHAP analysis. A total of 12 base learners were used, and the “stacking” model, which is a combination of the 6 best models, was tested on three different datasets. Both the accuracy and explainability of the model are highlighted. The authors achieved high metrics such as 99.28% accuracy, 99.55% precision, 97.56% recall, and 98.49% F1-score for three cancer types. While most studies focus on only one type of cancer, in this study, three different types were considered, which can be considered as a significant contribution. With SHAP analysis, the decision mechanisms of the model are made comprehensible, and two different stacking approaches are tested and compared with random and performance-selected models. The study achieved higher accuracy and AUC than other methods in the literature. However, I would kindly suggest the following suggestions to the authors.
- The lung cancer data contains only 309 samples, which is a problem in terms of generalizability, and I suggest that it be expanded.
Response: We acknowledge that the lung cancer dataset is relatively small, despite being upsampled to 537 samples. While a larger dataset could have been selected, we intentionally chose this smaller dataset to facilitate experimentation with varying sample sizes. In contrast, the cervical and breast cancer datasets are more extensive. Conducting experiments with a diverse array of datasets will enhance the model's generalizability.
- The data are taken from open sources, but geographical and demographic diversity is limited; this should be explained in detail, and this situation should be resolved.
Response: We sincerely appreciate your valuable feedback. We acknowledge the limitations in the geographical and demographic diversity of the datasets employed in our study. Our research utilized openly available datasets, which, while beneficial, may not comprehensively represent the diversity found in more extensive data sources. This limitation is addressed in the conclusion section. We greatly value your suggestion and will certainly explore ways to enhance the diversity of our data in future research endeavors. Thank you for bringing this matter to our attention.
- Only SMOTE is used, there is no comparison with different balancing techniques (ADASYN, Tomek Links), other comparisons should be included.
Response: We appreciate the importance of exploring various data-balancing techniques. While our manuscript primarily focuses on the use of SMOTE for addressing class imbalance, we would like to clarify that we also considered alternative methods, such as cost-sensitive learning and ensemble undersampling. However, we selected SMOTE due to its proven effectiveness in enhancing minority class representation, its widespread acceptance in medical datasets, and its alignment with previous studies (e.g., DOI: 10.1186/1471-2105-14-106, DOI: 10.62527/joiv.8.3.2283, DOI: 10.3390/diagnostics14232634). Given our emphasis on practical clinical applicability, SMOTE provided a well-validated and widely adopted solution for ensuring balanced data distribution while preserving model performance. This justification has been incorporated into the revised manuscript (Section 4).
- SHAP images are not detailed enough; also, clinical interpretation is missing.
Response: We sincerely appreciate your insightful comment, which highlights the crucial importance of both interpretability and clinical relevance in SHAP analysis, especially within the medical field. We would like to kindly point out that Subsection 6.2 of our manuscript provides a comprehensive account of the XAI-SHAP analysis, emphasizing the significant clinical importance of the top SHAP-identified features. Our dedication to clarity is reflected in the detailed sub-sections (6.2.1 and 6.2.2), which thoroughly address both global and local interpretations. For global interpretation, SHAP summary plots effectively illustrate the overall feature importance across datasets. Meanwhile, for local interpretation, SHAP waterfall and force plots are meticulously presented for individual predictions, each accompanied by comprehensive textual descriptions that explain the direction and magnitude of feature impacts. Moreover, we offer an extensive discussion of each SHAP visualization (Figures 24–32), providing clinical interpretations of the most influential features for each cancer type. For instance, in the case of lung cancer, features such as fatigue and alcohol consumption are rigorously analyzed in the context of established lifestyle-related risk factors. This comprehensive approach not only enhances understanding but also strengthens the clinical applicability of our findings.
- It was not tested with real-world data, and no evaluation of clinical integration was presented.
Response: We appreciate your feedback and acknowledge the significance of testing models with real-world data and evaluating clinical integration. Although our current study concentrated on theoretical frameworks and simulations using available datasets, we recognize that incorporating real-world data would enhance our findings. This limitation, regarding the absence of testing with real-world data and the evaluation of clinical integration, is noted in the conclusion section as a potential direction for future research. Nevertheless, we believe that our simulated study effectively demonstrates the model's efficacy. Simulations facilitate controlled experimentation and a comprehensive assessment of various scenarios, offering valuable insights into the model's performance. We value your understanding and the opportunity to clarify this point. Your insights are greatly appreciated and will inform our future work.
- Table 1 Evaluation metrics are well explained, Table 2 has a detailed comparison of 12 basic models; however, it is too dense and could have been better grouped.
Response: We are delighted to receive your positive feedback on Table 1 (now Table 2 in the revised manuscript). Regarding Table 2 (now Table 6), we understand its complexity. After carefully evaluating various alternatives, we believe that this version is the most concise and effective. While a graphical representation might seem appealing, it could have unnecessarily extended the length of our already comprehensive article. Instead, we have thoughtfully employed a colored heat map to enhance the visualization of the comparative performance of each model across datasets, ensuring clarity and impact without compromising the article's conciseness.
- Confusion matrices are presented in Table 5, but they should be supported with clear visualizations.
Response: We sincerely appreciate your suggestion. While we recognize that a figure representation of the confusion matrix could indeed enhance visual appeal, we have thoughtfully decided to present it in a table format. This choice is primarily motivated by our need to conserve space in our already extensive paper, which includes 32 figures. By opting for a table, we aim to ensure clarity and conciseness, allowing readers to focus on the critical insights without overwhelming them with additional figures. We hope you understand our reasoning and appreciate your understanding.
- Figure 4-9: Performance comparisons are detailed, but axis names should be written more clearly in some graphs. Figure 10-12: AUC-ROC curves are useful, but visual resolution could be improved.
Response: We sincerely appreciate your suggestion. While we understand the potential advantages of including axis names in the graphs, we made a considered decision to omit them to ensure the graphs fit seamlessly within the page size. Including axis names might affect the visual appeal, readability, and identifiability of the graphs. It is important to note that the Y-axis conventionally represents the values of the performance metrics of the models, a convention that is widely recognized. In most figures, data labels are already provided to enhance clarity. Regrettably, due to the graph's size, data labels could not be included in Fig. 4. Our approach aims to prioritize clarity and visual integrity, ensuring that the essential information is conveyed effectively.
We appreciate the reviewer's observation regarding the readability of Figures 10-12 in normal view. While these figures are program-generated and may have certain limitations, we kindly suggest utilizing the zoom function to enhance their clarity and detail, ensuring a more effective visualization.
- Figure 13-20: Comparisons for each metric are detailed. However, some figures repeat each other and should be simplified.
Response: We sincerely appreciate the reviewer's insights and understand the concern regarding potential redundancy. However, we would like to kindly emphasize that our strategy of presenting results from multiple analytical perspectives is both intentional and essential. The series of Figures 13 to 20 has been carefully crafted to achieve three important objectives:
- To provide a comprehensive comparison of performance metrics—such as accuracy, recall, precision, F1-score, AUC, etc.—for the top-performing models, including our innovative stacking models.
- To offer an in-depth analysis of each model's behavior across various cancer datasets, thereby providing invaluable insights.
- To facilitate a holistic comparative view of average performance across all diseases and evaluation metrics, thereby enhancing the robustness and reliability of our findings.
This multifaceted approach not only enriches the analysis but also strengthens the validity of our conclusions.
- The transferability of the model is unclear, and the applicability to different datasets or populations has not been tested.
Response: We sincerely appreciate the reviewer’s valuable concern regarding the model’s transferability. In response to this, we thoughtfully designed and evaluated our stacking ensemble approach using three distinct cancer datasets (lung, breast, and cervical) with varying sample sizes, feature dimensions, and clinical/lifestyle attributes. Specifically:
- Lung cancer: 309 samples, 15 features (lifestyle/clinical).
- Breast cancer: 569 samples, 30 features (imaging-derived).
- Cervical cancer: 835 samples, 35 features (clinical/histological).
By rigorously testing the model across these heterogeneous datasets—each representing various data sources, scales, and biomedical contexts—we were able to illustrate its adaptability to a range of cancer prediction tasks. The consistently high performance (e.g., average AUC > 0.99) indicates a commendable robustness to dataset variability, which is a significant indicator of its transferability.
We acknowledge that further validation on additional cancer types, such as prostate and colorectal, would indeed enhance the generalizability of our findings. However, we believe the current results indicate the model's potential applicability to: other cancers with similar tabular data, such as risk factors and clinical measurements, and new populations, provided the input features are consistent with our training domains.
- In particular, data preprocessing, missing data management, and feature selection should be presented in more detail.
Response: We sincerely appreciate your insightful feedback and would like to kindly offer some clarification on the following points:
Missing data and outliers: We have taken steps to address the issues of missing values and outlier detection across all datasets, as outlined in Section 4. Specifically, we have implemented imputation techniques to manage missing values and employed the IQR method to identify and rectify outliers.
Class balancing: We have expanded our explanation of the class balancing strategy by utilizing the SMOTE in Section 4. This includes a detailed discussion of its impact, on all datasets.
Feature Selection and Importance: For the analysis of feature contribution, we have utilized SHAP method, to evaluate both global and local feature importance. This analysis is thoroughly discussed in Section 6.2, with additional interpretations for each cancer type provided in Subsection 6.2.1.
- Code sharing is important for the reproducibility of the work. Also, optimization was done with high precision, but the recall was low. It may miss important cases in the real world.
Response: We sincerely appreciate the reviewer's suggestion. In response, we have made the complete code for this experiment available in a public repository GitHub: https://github.com/Shahid92-Phd/Multi-Cancer. Thank you for your valuable input. These comments substantially enhanced our manuscript.
Round 2
Reviewer 1 Report
Comments and Suggestions for Authors
Overall, the authors have improved the quality of the article. However, there are a few minor comments.
- In line 361, the figure number must be specified.
- In line 582, the period after the "s" must be removed.
- It would be interesting to know how many combinations of parameters the authors used in the grid search (how many combinations of parameters they considered)? How many combinations of parameters were there in the random search? Perhaps this would help explain the victory of the grid search over the random search.
Author Response
Overall, the authors have improved the quality of the article. However, there are a few minor comments.
- In line 361, the figure number must be specified.
Response: Thank you for the suggestion. The figure number is now mentioned.
- In line 582, the period after the "s" must be removed.
Response: Thanks for pointing out the mistake. It is now corrected. We appreciate your thorough observation.
- It would be interesting to know how many combinations of parameters the authors used in the grid search (how many combinations of parameters they considered)? How many combinations of parameters were there in the random search? Perhaps this would help explain the victory of the grid search over the random search.
Response:
We appreciate the reviewer’s insightful question regarding the scope of our hyperparameter search. Below, we provide the requested details.
For grid search, we constructed a parameter grid tailored to the nature and range of its hyperparameters for each algorithm. For instance, in the case of the NB classifier, eight key hyperparameters were considered, including random_state (binary), c (integer), gamma (decimal), kernel (linear, polynomial, RBF), probability (binary), verbose (binary), refit (binary), and verbose levels (5 discrete levels). To maintain computational feasibility, we limited the range of integer and decimal parameters to a maximum of 12 values each. This resulted in a comprehensive search space, such as 2 × 12 × 12 × 3 × 2 × 2 × 2 × 5 combinations for the NB model. This exhaustive pattern of evaluation was consistently applied across all other models included in the study, ensuring that possible interactions between hyperparameters were systematically explored.
In contrast, random search samples hyperparameter combinations stochastically from the same predefined parameter sets. In our experiments, we generated 12 random combinations for each model, allowing us to compare the efficiency and effectiveness of this approach with grid search. While random search offers greater computational efficiency and can be advantageous in high-dimensional spaces, its stochastic nature may overlook critical regions of the hyperparameter space that grid search is able to systematically evaluate.
This explanation is added in Section 5.2. Thank you once again for your valuable feedback.
Reviewer 2 Report
Comments and Suggestions for Authors
None, the authors replied to all comments.
P.S. About this comment, “Section 2. All bibliographical references were skipped.” In the previous version, all references ‘[XX]’ were not present in Section 2, probably there was a problem generating the pdf.
Author Response
None, the authors replied to all comments.
P.S. About this comment, “Section 2. All bibliographical references were skipped.” In the previous version, all references ‘[XX]’ were not present in Section 2, probably there was a problem generating the pdf.
Response: Thank you for your positive feedback and for confirming that we have addressed all comments. We also believe that the reference issue in Section 2 likely occurred during the PDF generation process. We appreciate your understanding.
Reviewer 3 Report
Comments and Suggestions for Authors
I appreciate the patience and hard work of the authors in this paper. I see that they have kindly complied with the suggestions made by me. In this sense, the paper is satisfactory to me and I kindly recommend its acceptance.
Comments on the Quality of English LanguageI appreciate the patience and hard work of the authors in this paper. I see that they have kindly complied with the suggestions made by me. In this sense, the paper is satisfactory to me and I kindly recommend its acceptance.
Author Response
I appreciate the patience and hard work of the authors in this paper. I see that they have kindly complied with the suggestions made by me. In this sense, the paper is satisfactory to me and I kindly recommend its acceptance.
Response: Thank you for your encouraging comments and for recognising our efforts to address your suggestions. We appreciate your patience and support throughout the review process. Your recommendation for acceptance is highly valued.